# STEP-AWARE POLICY OPTIMIZATION FOR REASONING IN DIFFUSION LARGE LANGUAGE MODELS

## ABSTRACT

Diffusion language models (dLLMs) offer a promising non-autoregressive paradigm for text generation, but training them for complex reasoning remains challenging. Current reinforcement learning approaches typically rely on sparse, outcome-based rewards, which can lead to inefficient exploration and "unstructured refinement", where the model's iterative denoising steps fail to contribute meaningfully to the solution. While Process Reward Models (PRMs) effectively mitigate similar issues in autoregressive models, they often require expensive human annotation or external verifiers. In this work, we propose Step-Aware Policy Optimization (SAPO), a method to derive automatic process rewards for dLLMs without external supervision. By leveraging the diffusion model's natural operation, we design a reward function that incentivizes distributing problem complexity evenly across the denoising trajectory. This intrinsic process supervision guides the model to learn structured, robust reasoning paths, reducing the risk of derailing from correct traces. Our empirical results demonstrate that SAPO significantly improves performance on challenging reasoning benchmarks and enhances the interpretability of the generation process.

## 1 INTRODUCTION

Diffusion large language models (dLLMs) have emerged as a compelling alternative to traditional autoregressive models (ARMs), offering the potential to significantly speed up model inference through their parallel, non-sequential generation process (Nie et al., 2025; Sahoo et al., 2024; Gong et al., 2024; Ye et al., 2025). In particular, mask-based dLLMs (MdLLMs) initialize a sequence with special token `[MASK]` and iteratively refine this sequence into coherent text. While this paradigm has shown promise on various general tasks, effectively training MdLLMs for complex, multi-step reasoning remains a significant challenge.

In the realm of autoregressive models, Process Reward Models (PRMs) (Uesato et al., 2022; Lightman et al., 2023) have become an effective solution for improving reasoning. By providing dense, step-by-step supervision rather than a single sparse reward at the end, PRMs encourage models to maintain coherent reasoning throughout the generation. However, obtaining such dense supervision is costly, often requiring large-scale human annotation or external verifiers. Consequently, current reinforcement learning (RL) methods for MdLLMs, such as GRPO (Shao et al., 2024) adapted in diffu-GRPO (Zhao et al., 2025), typically rely solely on sparse, outcome-based rewards.

This reliance on sparse rewards can be problematic. Without intermediate guidance, models are prone to what we term *unstructured refinement*. While models may maintain local textual coherence, they often fail to utilize the iterative denoising process for logical progression. This results in the model wasting steps on unproductive tokens—manifesting as repetitive loops (mode collapse) or coherent but vacuous 'fluff'—forcing the final few steps to bridge the entire complexity gap. This not only inefficiently uses the diffusion process but also increases the risk of generating hallucinations or inconsistent reasoning paths that only coincidentally arrive at the correct answer (Figure 1).

To address this gap, we propose **Step-Aware Policy Optimization (SAPO)**, an algorithm that extracts automatic process rewards for dLLMs. Our key insight is that we can leverage the unique, inherent iterative structure of diffusion models to provide this supervision without external costs. We introduce a method to estimate the contribution of specific denoising intervals by comparing the expected outcome of intermediate states. This allows us to reward denoising steps that demonstrably

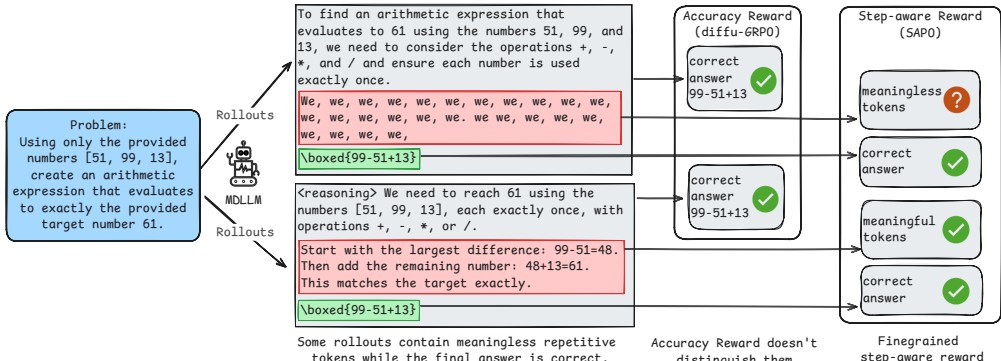

Figure 1: The problem of *unstructured refinement*. A standard MdLLM trained with outcome-only rewards produces a correct answer but fills its reasoning trace with meaningless tokens. While this specific example exhibits mode collapse (a coherence failure), it serves as a stark illustration of a broader issue: the iterative process is not incentivized to reduce problem complexity, allowing the model to 'spin' on unproductive steps while coincidentally hitting the correct answer.

reduce the remaining problem complexity, providing a dense supervision signal entirely from within the model's own rollouts.

This approach serves as a self-supervised mechanism to align the diffusion process with effective reasoning structures. By incentivizing incremental progress, we encourage the model to distribute the problem-solving load across the entire generation process, mitigating the risks associated with unstructured refinement.

Our contributions are as follows:

1. We identify the lack of process supervision as a key limitation in existing MdLLM training, leading to inefficient and potentially unstable reasoning processes.
2. We introduce SAPO, a novel RL framework that provides automatic process rewards for dLLMs. It leverages the diffusion model's natural operation to incentivize progressive complexity reduction without needing external reward models or verifiers.
3. We demonstrate empirically that SAPO leads to significant improvements in both final performance and the quality of generated reasoning paths across multiple benchmarks, validating the effectiveness of intrinsic process supervision.

## 2 RELATED WORK

**Mask-based diffusion-based large language models.** LLaDA (Nie et al., 2025) proposes a mask-based diffusion-based large language model (dLLMs). It gradually removes the mask token in each diffusion step. Based on LLaDA, diffu-GRPO (Zhao et al., 2025) assumes the generated tokens are independent and proposes a randomly masked prompt to estimate the token probability for reinforcement learning with diffusion models. WINO (Hong et al., 2025) proposes a training-free sampling strategy to use a low confidence threshold to generate a draft response and use a high threshold for second verification. TSE (Wang et al., 2025c) observes that the answers generated in intermediate diffusion steps can also be correct and therefore proposes a weighted voting strategy to get the final answer. ReMDM (Wang et al., 2025a) proposes a remasking sampler to address the problem that the generated tokens in dLLMs cannot be revoked. wd1 (Tang et al., 2025) proposes a weighted likelihood estimation for the sequence. Many approaches have been proposed to improve the efficiency of dLLMs, such as KV-cache (Wu et al., 2025; Song et al., 2025; Liu et al., 2025b; Ma et al., 2025). MDLM (Sahoo et al., 2024) derives a continuous-time, Rao-Blackwellized objective for training mask-based dLLM. LongLLaDA (Liu et al., 2025a) proposes an NTK-based RoPE extrapolation to allow long-context text generation. DiffuCoder (Gong et al., 2025) proposes a coupled sampling scheme to estimate the likelihood for GRPO training. MDPO (He et al., 2025) introduces a running confidence remasking strategy to allow low-confidence tokens to be remasked again during inference time.

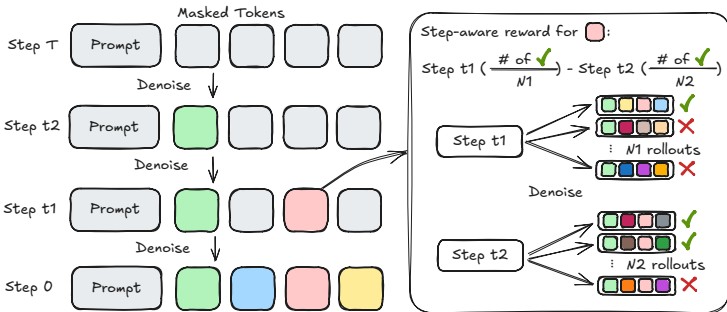

Figure 2: Illustration of the proposed step-aware reward. To encourage intermediate generations to contribute meaningfully to the final outcome, we generate new rollouts from randomly selected steps $t_1, t_2$ and estimate their contribution by the difference in outcome rewards. A larger difference indicates a higher contribution toward the final correct answer.

**Process reward model.** Verification models have been shown to improve the multi-step reasoning ability of LLMs. Unlike the outcome verifier (Cobbe et al., 2021; Yu et al., 2023) which examines the correctness of the final outcome, the process reward models enhance feedback accuracy by identifying and localizing errors within generated responses. However, collecting step-wise feedback can be costly, especially with human annotators (Uesato et al., 2022; Lightman et al., 2023). Therefore, many efforts have been devoted to the automatic extraction of process rewards. One standard way to assess process correctness is by estimating, via Monte Carlo (MC) methods, the empirical probability of reaching the correct final answers. Given an intermediate step of reasoning, MATH-SHEPHERD (Wang et al., 2023) asks completers to finalize multiple subsequent reasoning processes and estimate the potential of this step based on the correctness of all decoded answers. (Luo et al., 2024) proposes a Monte Carlo Tree Search algorithm to identify the first error in the reasoning process. (Zhang et al., 2025) argues that the MC-based estimation can be noisy and requires an additional LLM-as-judge to filter the process reward data. Inspired by (Wang et al., 2023), (Wang et al., 2025b) constructs process rewards for multi-modal LLMs. (Zhang et al., 2024a) proposes a tree search policy with process rewards. Implicit process rewards (Yuan et al., 2024; Cui et al., 2025) trains the outcome reward model and can obtain the token-level process reward as log-likelihood ratios of the policy and reference models.

## 3 STEP-AWARE POLICY OPTIMIZATION FOR STRUCTURED REASONING

Our primary goal is to mitigate the risk of inefficient or derailed reasoning in MdLLMs. We achieve this by providing dense, process-level supervision that encourages the model to distribute problem complexity evenly across generation steps. To obtain this supervision without external annotators or reward models, we leverage the diffusion model's own iterative nature.

We introduce Step-Aware Policy Optimization (SAPO), a reinforcement learning framework built upon Group Relative Policy Optimization (GRPO) specifically adapted for MdLLMs. Its core innovation is a novel, automatic process-based reward function.

### 3.1 PRELIMINARY: GROUP RELATIVE POLICY OPTIMIZATION (GRPO)

GRPO is a powerful on-policy algorithm for enhancing the capabilities of language models (Shao et al., 2024). We adapt it for the MdLLM setting.

**Response sampling.** Given a question $\mathbf{Q}$, we use the current policy $\pi_\theta$ to generate $G$ candidate responses $\{\mathbf{R}^{(1)}, \mathbf{R}^{(2)}, \ldots, \mathbf{R}^{(G)}\}$. Each response $\mathbf{R}^{(i)}$ is assigned a reward $r_i$, based on the correctness of the final answer. From these, we can compute a mean-normalized advantage for each response, $A_i = r_i - \text{mean}(\{r_j\}_{j=1}^G)$. This advantage is distributed across all tokens in the response.

**Learning objective.** The optimization follows the standard proximity policy optimization (PPO) (Schulman et al., 2017)-style clipped objective for stable updates, regularized by a KL-

divergence term against a reference policy $\pi_{\text{ref}}$:

$$\mathcal{L}_{\text{GRPO}}(\theta) = \mathbb{E}_{\mathbf{Q}\sim\mathcal{D},\, \mathbf{R}^{(1)},...,\mathbf{R}^{(G)}\sim\pi_\theta(\cdot|\mathbf{Q})} \left[ \frac{1}{G}\sum_{i=1}^{G}\frac{1}{|\mathbf{R}^{(i)}|}\sum_{k=1}^{|\mathbf{R}^{(i)}|} \min\left(\rho_i^k A_i,\ \text{clip}(\rho_i^k, 1-\varepsilon, 1+\varepsilon)\, A_i\right) \right.$$

$$\left. - \beta\, D_{\text{KL}}[\pi_\theta(\cdot\mid\mathbf{Q})\,\|\,\pi_{\text{ref}}(\cdot\mid\mathbf{Q})] \right],$$

(1)

where the likelihood ratio for the $k$-th token of response $\mathbf{R}^{(i)}$ is $\rho_i^k = \frac{\pi_\theta(\mathbf{R}^{(i),k}|\mathbf{Q},\mathbf{R}^{(i),<k})}{\pi_{\theta_{\text{old}}}(\mathbf{R}^{(i),k}|\mathbf{Q},\mathbf{R}^{(i),<k})}$. A key challenge in applying this to MdLLMs is estimating the sequence likelihood $\pi_\theta(\mathbf{R}^{(i)}\mid\mathbf{Q})$, which we address with existing techniques (Zhao et al., 2025; Gong et al., 2025; Tang et al., 2025).

## 3.2 STEP-AWARE STRUCTURED REFINEMENT FOR MdLLMS

Standard GRPO for MdLLMs defines the advantage $A_i$ based solely on outcome-based rewards (e.g., final answer accuracy). This may lead to unstructured refinement, as it can equally reinforce responses that are correct by chance despite having flawed reasoning as illustrated in Fig.1. To enforce a structured reasoning process, we introduce a step-aware reward.

**Denoising steps in MdLLMs.** Given an input question $\mathbf{Q}$, MdLLM begins by preparing a sequence of mask tokens [mask] of pre-defined length and initializing the denoising process at step $t = T$. At each iteration, the model receives the partially masked sequence and incrementally replaces mask tokens with decoded tokens according to a chosen decoding strategy (e.g., decoding only those tokens whose confidence exceeds a specified threshold). At an intermediate step $t$, the sequence thus contains a mixture of text and mask tokens, such as "an apple [mask] [mask] is on the [mask]". When $t = 0$, all mask tokens [mask] are fully resolved into text tokens.

**Evaluating denoising steps with step-aware reward.** To encourage structured reasoning within the denoising process, one possible approach is to manually annotate intermediate generations, following the methodology of process reward models developed for ARMs (Uesato et al., 2022). However, unlike ARMs, where tokens are decoded sequentially and intermediate outputs are inherently structured and separable, annotating intermediate states in MdLLMs poses additional challenges. This difficulty arises because MdLLM intermediate generations consist of a mixture of text and mask tokens, often arranged in a non-deterministic order due to the parallel decoding mechanism. For instance, an intermediate state might appear as "an apple [mask] [mask] is on the [mask]", where incomplete decoding obscures clear annotation.

To address this challenge, we propose measuring the incremental progress achieved between different stages of the denoising process. Specifically, we randomly sample two denoising timesteps, $t_1$ and $t_2$, such that $0 \leq t_1 < t_2 \leq T$. Let $x_{t_1}$ and $x_{t_2}$ denote the intermediate generations at these steps. To evaluate the contribution of the denoising steps between $t_2$ and $t_1$, we generate full response rollouts from each state, yielding $\{\mathbf{R}^{(j)}(x_{t_1})\}_{j=1}^{N_1}$ and $\{\mathbf{R}^{(j)}(x_{t_2})\}_{j=1}^{N_2}$.

The step-aware reward is defined as the difference in the expected outcome rewards:

$$R_{\text{process}}(t_1, t_2) = \frac{1}{N_1}\sum_{j=1}^{N_1}\mathbf{1}[\mathbf{R}^{(j)}(x_{t_1})] - \frac{1}{N_2}\sum_{j=1}^{N_2}\mathbf{1}[\mathbf{R}^{(j)}(x_{t_2})], \qquad (2)$$

where $\mathbf{1}[\cdot]$ denotes an indicator function that evaluates the correctness of the final response. A positive value of $R_{\text{process}}$ indicates that the denoising steps between $t_2$ and $t_1$ made a meaningful contribution, thereby reducing its complexity. Importantly, this formulation eliminates the need to manually annotate intermediate diffusion states or to design task-specific process reward models.

**Efficient reward estimation.** Although MdLLMs offer faster inference compared to ARMs, generating multiple responses from intermediate states at two different timesteps can still be computationally expensive. To mitigate this cost, we focus on an important special case where $t_2 = T$. At this point, the intermediate generation $x_{t_2}$ consists entirely of mask tokens [mask] ... [mask]. Consequently, the second term $\frac{1}{N_2}\sum_{j=1}^{N_2}\mathbf{1}[\mathbf{R}^{(j)}(x_{t_2})]$ in Eq. 2 corresponds to the model's accuracy when conditioned solely on the input question prompt $\mathbf{Q}$. In this case, we set $t_2 = T$,

**High-Complexity
Unstructured Refinement**

**Low-Complexity
Structured Decomposition**

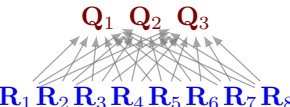

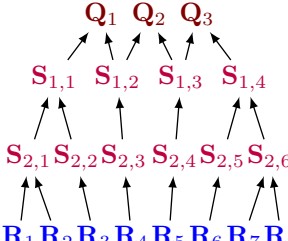

Figure 3: **Complexity reduction via structured decomposition.** (*Left*) A model without process supervision attempts to solve the complex mapping from Question $\mathbf{Q}$ to Response $\mathbf{R}$ directly, often leading to difficulty bottlenecks. (*Right*) A model guided by process rewards decomposes the problem into intermediate latent states $\mathbf{S}$, ensuring each step performs a small, manageable reduction in complexity (sparsity).

$\mathbf{R}^{(j)}(x_{t_2}) = \mathbf{R}^{(j)}$, and $N_2 = G$. Since full response rollouts are already available from GRPO-based accuracy reward computation, this term can be directly estimated without additional inference. In other words, we substitute the mean accuracy reward as a surrogate for the second term, effectively halving the inference cost required to compute $R_{\text{process}}(t_1, t_2)$.

In principle, the full trajectory reward could be computed by evaluating all denoising steps from $T$ down to $0$, but this approach would incur prohibitively high computational cost. Instead, we find that estimating the reward using a randomly sampled interval $(t_1, t_2)$ serves as an effective and efficient approximation of the overall reasoning process during generation. Accordingly, we define

$$R_{\text{process}} := R_{\text{process}}(t_1, t_2). \tag{3}$$

**Up-weighted advantage computation.** In GRPO (Shao et al., 2024) and diffu-GRPO (Zhao et al., 2025), the advantage is computed by normalizing all rewards across rollouts for a given input prompt. However, in our preliminary experiments, we observe that directly applying such normalization to the step-aware reward can degrade model performance. This occurs because samples with $R_{\text{process}} = 0$ are pushed further away during mean-normalization, yielding negative advantages. Such treatment is suboptimal, as these samples (with correct answers and flawed reasoning steps) may still contribute positively to model learning. To address this issue, we introduce an up-weighted strategy for computing the total advantage of response $\mathbf{R}^{(i)}$:

$$A_i^{\text{total}} = A_i + \mathbf{1}[A_i > 0] \cdot R_{\text{process}} \tag{4}$$

where $A_i$ is the advantage for response $\mathbf{R}^{(i)}$. Crucially, up-weighting is applied only to responses that both yield a correct final answer and already possess a positive advantage. This design ensures that we reinforce valid reasoning paths without rewarding intermediate progress that ultimately leads to incorrect solutions, and without penalizing correct answers that may contain imperfect reasoning.

This composite advantage thus integrates correctness with structured, productive reasoning, directly incentivizing the model to adhere to the principle of hierarchical decomposition.

### 3.3 THEORETICAL UNDERSTANDING: COMPLEXITY DISTRIBUTION

To ground our method in a formal framework, we interpret the benefits of intermediate rewards through the lens of *complexity reduction*. A reasoning task defines a high-complexity constraint between a question $\mathbf{Q}$ and a response $\mathbf{R}$. Directly generating $\mathbf{R}$ that satisfies $\mathbf{Q}$ without structured guidance is difficult because the search space is vast and the dependency is complex (Figure 3, Left). We do not posit this hierarchical structure as a rigid, universal cognitive model. Rather, we propose the hierarchy as a flexible abstraction for the potential reasoning complexity. Simpler problems activate only a sparse subgraph of the available constraints. In the context of diffusion, this manifests as trivial transformations where the 'reasoning' happens implicitly via smooth constraint satisfaction in the latent space, without requiring complex structural decomposition.

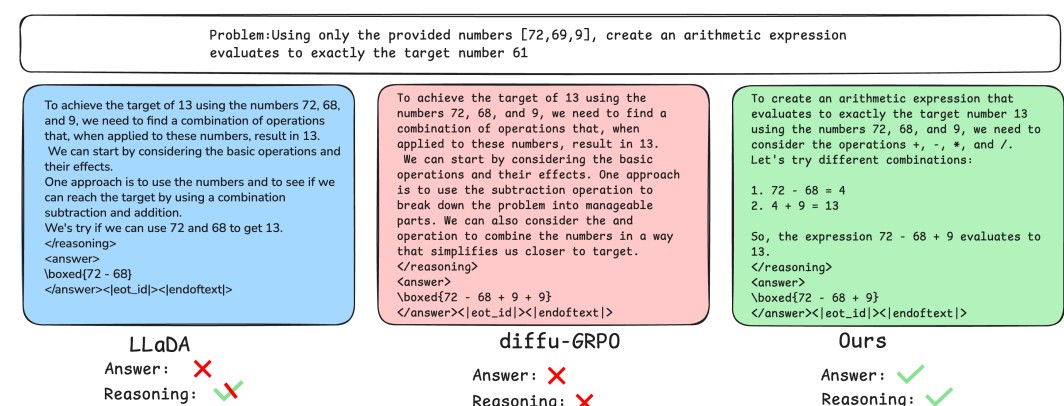

Figure 4: Comparison of generated responses across models. LLaDA (Nie et al., 2025) and diffu-GRPO (Zhao et al., 2025) both produce incorrect answers to the evaluation question. LLaDA's response includes a brief but partially meaningful reasoning step toward the end, whereas diffu-GRPO continues generating verbose sentences that contribute little to the final prediction. In contrast, our model provides a structured reasoning process and successfully arrives at the correct answer. This highlights that optimizing solely for accuracy-based rewards may lead to sub-optimal outcomes, as such rewards overlook the quality and coherence of reasoning within the response.

Standard dLLM training paradigms are often agnostic to intermediate progress, suffering from **unstructured refinement**: the model may waste early steps on irrelevant tokens or derail from a logical path, forcing it to bridge a massive complexity gap in the final few steps.

Ideally, the difficulty of the problem should decrease monotonically and gradually as the diffusion proceeds. We formally characterize this as a **sparsity constraint** on the latent reasoning process (see Appendix B for details). Intuitively, if a model can decompose a complex function into a composition of sparse, simple functions, it can more easily learn a natural, robust reasoning process.

**Theorem 3.1** (Informal: Complexity Distribution). *A reasoning model that distributes the computational load (e.g., satisfies a **sparsity constraint**), where each transition resolves only a limited subset of dependencies, learns a natural, robust reasoning process that is less prone to unstructured refinement.*

Our proposed method, SAPO, directly operationalizes this insight. By rewarding intervals that show a measurable increase in the probability of correctness, we encourage the model to distribute the complexity reduction evenly across all steps, ensuring that every stage of the diffusion process contributes a small, manageable piece of the solution.

## 4 EXPERIMENTS

### 4.1 SETUP

We build our model on top of diffu-GRPO (Zhao et al., 2025) and adopt the same experimental setup unless otherwise specified. We provide implementation details in the Appendix.C.

**Datasets.** We evaluate on four benchmarks: (1) GSM8K (Cobbe et al., 2021), using 7,374 training and 1,319 test problems; (2) MATH (Lightman et al., 2023), with 7,500 training and 500 test problems; (3) COUNTDOWN, a synthetic dataset of 490K training and 256 test samples requiring arithmetic expression generation; and (4) SUDOKU, $4 \times 4$ puzzles evaluated on a 256-sample split.

**Baselines.** We compare against recent state-of-the-art MdLLMs: LLaDA (Nie et al., 2025), Diffu-GRPO (Zhao et al., 2025), TSE (Wang et al., 2025c), and WINO (Hong et al., 2025), as well as models further fine-tuned on the reasoning dataset s1K (Muennighoff et al., 2025).

| Model / Seq Len | COUNTDOWN | | | GSM8K | | | SUDOKU | | | MATH | | |
|---|---|---|---|---|---|---|---|---|---|---|---|---|
| | 128 | 256 | 512 | 128 | 256 | 512 | 128 | 256 | 512 | 128 | 256 | 512 |
| LLaDA | 20.7 | 19.5 | 16.0 | 68.7 | 76.7 | 78.2 | 11.7 | 6.7 | 5.5 | 26.0 | 32.4 | 36.2 |
| diffu-GRPO | 33.2 | 31.3 | 37.1 | 72.6 | 79.8 | 81.9 | 18.4 | 12.9 | 11.0 | **33.2** | 37.2 | 39.2 |
| TSE-Vote | 25.0 | 23.4 | 16.4 | 70.1 | 78.7 | 78.9 | × | × | × | 28.4 | 35.6 | 36.2 |
| WINO | - | 33.2 | - | - | 75.8 | - | - | 15.2 | - | - | 34.2 | - |
| SFT | 20.3 | 14.5 | 23.8 | 66.5 | 78.8 | 81.1 | 16.5 | 8.5 | 4.6 | 26.2 | 32.6 | 34.8 |
| SFT + diffu-GRPO | 34.8 | 32.0 | 42.2 | **73.2** | 81.1 | 82.1 | 22.1 | 16.7 | 9.5 | **33.8** | 38.6 | 40.2 |
| SFT + TSE-Reward | 41.5 | 42.6 | 54.7 | 72.1 | 80.0 | **83.0** | × | × | × | 31.2 | 35.4 | **41.4** |
| **diffu-GRPO+PRM** | - | - | - | 71.7 | 80.9 | 81.5 | - | - | - | 30.8 | 36.0 | 36.0 |
| **Ours** | **51.6** | **52.0** | **56.3** | 72.9 | **82.2** | 82.4 | **22.4** | **20.3** | **16.1** | 32.0 | **40.0** | 38.4 |

Table 1: Performance comparison on COUNTDOWN, GSM8K, SUDOKU, and MATH at different sequence lengths. "–" denotes unreported results; "×" denotes unsupported tasks. Without additional SFT on the reasoning dataset s1K (Muennighoff et al., 2025), our method achieves superior performance across all four tasks.

| Model/Seq Len | COUNTDOWN | | | sec/it |
|---|---|---|---|---|
| | 128 | 256 | 512 | |
| diffu-GRPO | 33.2 | 31.3 | 37.1 | 3.19 |
| diffu-GRPO+PRM | - | - | - | 7.58 |
| Ours-NoUpweight | 41.0 | 41.4 | 50.4 | 3.42 |
| Ours-Cover | 55.1 | 59.4 | 58.2 | 6.23 |
| Ours-Random | 55.4 | 54.7 | 59.8 | 4.76 |
| Ours | 51.6 | 52.0 | 56.3 | 3.42 |

Table 2: Ablation on different designs and effiency comparisons.

| Dataset | diffu-GRPO | Ours |
|---|---|---|
| COUNTDOWN | 4.37±2.41 | 3.80±2.04 |
| GSM8K | 2.37±0.80 | 2.12±0.74 |
| SUDOKU | 4.31±2.95 | 3.90±2.44 |
| MATH | 3.19±1.21 | 3.11±1.24 |

Table 3: The number of causal links across timesteps. With the proposed reward, our approach learns a sparser hierarchy (smaller mean), and the changes across timesteps (hierarchy levels) are smoother and more stable, as indicated by the smaller standard deviation.

## 4.2 RESULTS

**Alignment of reasoning process and final answer.** To assess how well MDLLMs produce intermediate reasoning that is consistent with the final answer, we analyze the alignment between the reasoning process and the output. Specifically, we input generations from LLaDA (Nie et al., 2025), diffu-GRPO (Zhao et al., 2025), and our model into GPT-5, asking it to evaluate "whether a user can reach the final answer by following the reasoning step by step." Results on the COUNTDOWN and GSM8K datasets are shown in Fig. 6. Our method achieves substantially higher alignment ratios across both datasets. This large improvement helps explain the performance gains in Table 1, as our proposed reward explicitly encourages the model to maintain consistency between reasoning steps and final answers through the diffusion-based generation process. We also provide example outputs from the three models in Fig. 1. As shown, LLaDA and diffu-GRPO generate less meaningful reasoning in their responses and ultimately produce incorrect answers.

**Comparison with using pretrained PRM**. Since our approach is fundamentally built upon the idea of process rewards, it is crucial to understand how it compares to an existing and widely adopted paradigm for process-level supervision: using a pretrained Process Reward Model (PRM) as the reward function. To this end, we adopted the pretrained Mistral-7B PRM checkpoint from Zhang et al. (2024b). We inserted their reasoning-step tags every 16 timesteps during masked-token decoding, fed the entire sequence into the PRM, and computed the process reward as the average PRM score across timestep intervals.

Despite PRMs being effective for test-time selection, we encountered several significant challenges when attempting to use them as training-time rewards for dLLM policy optimization as shown in Fig.5: (1) Huge memory consumption. Unlike our approach, which reuses the training model itself to compute rewards, the pretrained PRM introduces substantial GPU and CPU memory overhead, leading to much slower training (7.62 sec/it vs. 3.42 sec/it for ours; see Table 2). (2) Instability. Generated responses frequently caused the PRM to output NaNs, likely because it expects strictly

| Model | Training | SVAMP | ARC |
|---|---|---|---|
| LLaDA | - | 83.3 | 90.2 |
| diffu-GRPO | GSM8K | 83.0 | 89.8 |
| Ours | GSM8K | 84.0 | 90.2 |
| diffu-GRPO | MATH | 83.7 | 91.8 |
| Ours | MATH | 85.7 | **93.0** |
| diffu-GRPO | COUNTDOWN | 84.0 | 90.6 |
| Ours | COUNTDOWN | 84.0 | 87.5 |
| diffu-GRPO | SUDOKU | 85.0 | 91.0 |
| Ours | SUDOKU | **86.7** | 90.6 |

Table 4: Generalization ability comparison. The trained models are evaluated on unseen datasets: the reasoning benchmark SVAMP (Patel et al., 2021) and the commonsense benchmark ARC (Clark et al., 2018).

| Step | COUNTDOWN | | GSM8K | |
|---|---|---|---|---|
| | diffu-GRPO | Ours | diffu-GRPO | Ours |
| 1 | **1.56** | 1.17 | 12.81 | **16.53** |
| 8 | **2.73** | 1.56 | 9.48 | **16.91** |
| 16 | **3.12** | 2.34 | 13.04 | **19.33** |
| 24 | **4.69** | 1.95 | 17.21 | **21.61** |
| 32 | 6.64 | **27.34** | 24.26 | **30.86** |
| 40 | 12.50 | **33.98** | 39.27 | **41.17** |
| 48 | 19.53 | **37.11** | 49.96 | **50.57** |
| 64 | 33.2 | **51.6** | 72.6 | **72.9** |

Table 5: Accuracy of intermediate answers with sequence length 128 and 64 diffusion steps. Intermediate answers are obtained by decoding normally up to a target step and then decoding all remaining tokens in one pass.

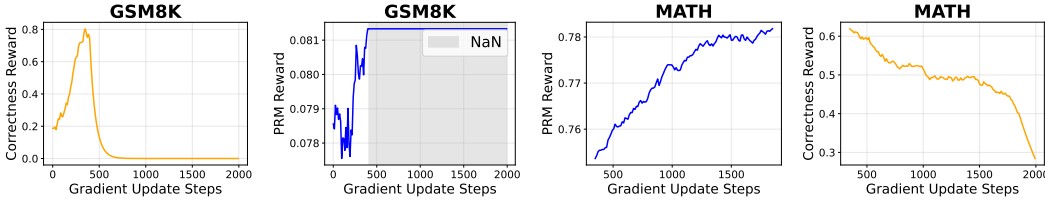

Figure 5: Two failure runs when using a pretrained PRM to assign rewards: instability and potential reward hacking. Unlike our rule-based reward, the PRM must process the full model-generated response through a large pretrained network. As a result, it often encounters unseen or irregular response formats, which can lead to numerical instabilities and NaN outputs. In addition, as shown on the right for the MATH dataset, although the PRM reward steadily increases during training, the accuracy reward actually decreases. This divergence suggests that the model learns to exploit weaknesses in the PRM scoring function—effectively hacking the reward model rather than improving its reasoning quality.

formatted inputs (e.g., explicit "step1/step2" markers). This forced us to replace NaN scores with zero, introducing further noise into the learning process. (3) Reward hacking. Although the PRM reward steadily increased during training, the actual task performance did not improve. This suggests that the policy model learns to exploit flaws in the PRM scoring function rather than improving its reasoning quality—a well-known failure mode for reward-model-based optimization. As shown in Table 1, diffu-GRPO+PRM achieves 71.7, 80.9, and 81.5 on GSM8K, while diffu-GRPO obtains 72.6, 79.8, and 81.9. Our method further improves to 72.9, 82.2, and 82.4. These results underscore that, even when compared against a strong pretrained PRM, our rule-based reward provides more stable optimization and better downstream performance, reinforcing the motivation for our design.

**Ablation study on the model design.** We now examine several design choices in our framework. Ours-NoUpweight removes the up-weighting strategy and applies the reward to all samples rather than only those with positive advantages. Ours-Cover computes the proposed reward across all timestep intervals, corresponding to the exact empirical average. Ours-Random selects $t_2$ uniformly at random instead of fixing $t_2 = T$. The results on the COUNTDOWN dataset are shown in Table 2.

The ablation results demonstrate the contributions of each component. Removing the up-weighting strategy (Ours-NoUpweight) already yields notable improvements over diffu-GRPO (e.g., 41.0 vs. 33.2 at sequence length 128), indicating that the reward formulation alone provides a substantial benefit. Computing rewards across all timestep intervals (Ours-Cover) achieves the strongest overall performance (55.1, 59.4, 58.2), but requires roughly twice the computation time (6.23 sec/it). Sampling $t_2$ at random (Ours-Random) achieves similarly strong accuracy (55.4, 54.7, 59.8), though it is still slower than our approach (4.76 sec/it), as it requires an additional forward pass and cannot reuse the final completion for filtering. Our full method (Ours) achieves performance close to Ours-Cover

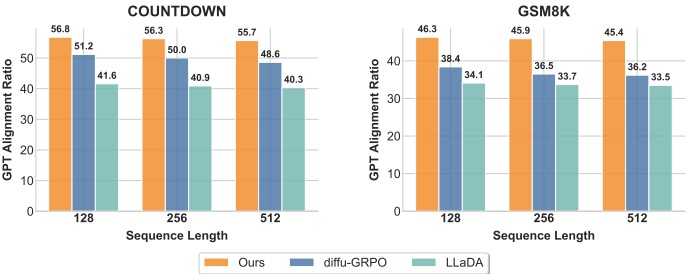
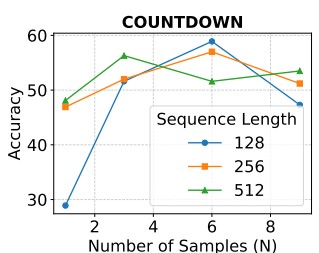

Figure 6: Model reasoning–outcome alignment ratio.

Figure 7: Ablation study.

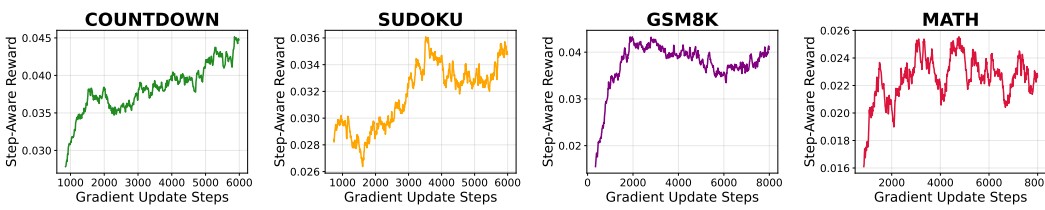

Figure 8: The training curve of our proposed step-aware reward.

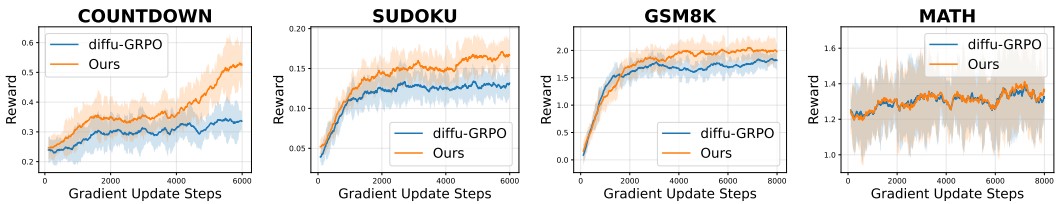

Figure 9: Reward curves during GRPO training. For fair comparison, we exclude our step-aware reward. The step-aware reward emphasizes responses that contain both the correct answer and meaningful reasoning, which in turn enhances the accuracy reward.

and Ours-Random (51.6, 52.0, 56.3), while maintaining a training speed comparable to diffu-GRPO (3.42 sec/it vs. 3.19 sec/it). This indicates that our design provides an efficient approximation of the full reward with minimal computational overhead. For completeness, we also report the runtime of diffu-GRPO+PRM (7.58 sec/it), which is substantially slower due to the additional memory and computation required by the pretrained PRM model.

**Our reward enables a more sparse and stable reasoning hierarchy.** Our theoretical analysis suggests that enforcing suitable sparsity constraints should lead the model to learn a more structured and well-organized reasoning hierarchy. To empirically verify this prediction, we examine the causal structure induced by the learned models. At each prediction step, we estimate the causal links between hierarchy levels using token prediction probabilities. For example, given a partially decoded sequence such as "I am [mask]", we count how many vocabulary tokens have prediction probability greater than a small threshold (e.g., 0.01), treating each such token as an active causal link. We then compute the average and standard deviation of the number of causal links across timesteps.

As shown in Table 3, our approach consistently produces fewer causal links than diffu-GRPO, indicating a sparser learned hierarchy. For instance, on COUNTDOWN, the number of causal links decreases from 4.37 to 3.80 (a 13.1% reduction), and on SUDOKU it decreases from 4.31 to 3.90 (a 9.5% reduction). Likewise, our model shows smaller standard deviations—e.g., from 2.41 to 2.04 on COUNTDOWN—demonstrating that the changes across hierarchy levels are smoother and more stable. Similar trends hold across GSM8K and MATH. These empirical findings align with our theoretical motivation and confirm that our reward encourages a more structured reasoning process.

**Superior benchmark performance.** Table 1 reports results on benchmarks: GSM8K, MATH, COUNTDOWN, and SUDOKU. Our approach outperforms baselines across most datasets and even surpasses those fine-tuned with additional reasoning dataset s1K (Muennighoff et al., 2025).

**The proposed reward is effectively learned and facilitates training.** We visualize the training rewards of diffu-GRPO (Zhao et al., 2025) and our model in Fig. 9. Our method consistently achieves higher total rewards—which combine both accuracy and format rewards—explaining the substantial performance gains observed on these datasets. For MATH, however, the training rewards of both methods remain similar. A similar phenomenon was also reported in (Zhao et al., 2025), where the diffu-GRPO model after SFT on s1k dataset (Muennighoff et al., 2025) attained rewards similar to those without SFT. We hypothesize that the MATH500 problems may be too challenging for the 8B base model and may be addressed with a larger dLLM. We further present the reward training curves in Fig. 8. The upward trend indicates that the model learns to favor responses yielding correct answers while adhering to reasoning processes that support the final outcome.

**Our model demonstrates strong generalization ability.** To thoroughly examine the proposed framework, we further evaluate our models on two unseen datasets: SVAMP (Patel et al., 2021) and ARC (Clark et al., 2018). The SVAMP dataset consists of numerous mathematical reasoning problems, while the ARC dataset focuses on commonsense reasoning tasks (e.g., "When oxygen combines with hydrogen, which substance is formed?"), where the model must select the correct answer from multiple choices. Notably, ARC is fundamentally different from our training datasets (e.g., GSM8K). Our model noticeably improves performance on both SVAMP and ARC.

**Our method enables further acceleration through higher intermediate accuracy.** Accelerating MDLLMs has been an active area of research (Li et al., 2025; Hong et al., 2025; He et al., 2025). Many approaches rely on the quality of intermediate responses: if these responses are accurate and contribute meaningfully to the final answer, generation can be accelerated. For instance, Prophet (Li et al., 2025) decides whether to decode all remaining tokens in a single step. Motivated by this, we analyze the accuracy of intermediate responses produced by our method. Specifically, during diffusion denoising, at each step, we additionally generate an answer by unmasking all remaining tokens at once. This gives us intermediate answers at every step, in addition to the final output obtained from fully decoding the masked sequence. We present results in Table.5. Across both datasets, our method achieves higher intermediate accuracy, suggesting that it may offer advantages over diffu-GRPO (Zhao et al., 2025) when combined with MdLLM acceleration techniques.

**Effect of the number of samples on the reward.** Our step-aware reward function is based on an averaged estimation of the accuracy of generated responses. To assess its reliability, we perform an ablation study by varying the number of samples, $N \in \{1, 3, 6, 9\}$. As illustrated in Fig. 7, when $N = 1$, the estimation becomes noisy and leads to suboptimal performance. In contrast, when $N \geq 3$, we observe substantial improvements over both baseline methods, LLaDA (Nie et al., 2025) and diffu-GRPO (Zhao et al., 2025), across sequence lengths of 128, 256, and 512. These results highlight the robustness of our proposed reward under different sampling configurations.

## 5 CONCLUSION AND LIMITATIONS

We address the challenge of training diffusion language models for complex reasoning, identifying the lack of process supervision as a key limitation that leads to *unstructured refinement*. To overcome this without incurring the high costs of external verifiers or human annotation, we introduce SAPO, an RL framework that derives automatic process rewards from the diffusion model's inherent iterative structure. Supported by the theoretical insight of progressive complexity reduction, SAPO incentivizes the model to distribute problem difficulty evenly across the denoising trajectory, fostering structured and robust reasoning. Our empirical results demonstrate that SAPO significantly improves performance on challenging reasoning benchmarks and enhances the coherence of the generation process.

**Limitation.** Our method relies on the mean-field assumption used in diffu-GRPO (Zhao et al., 2025) to estimate the log-likelihood of generated responses, which inherently neglects token-level dependencies. Unfortunately, this assumption is difficult to remove because, unlike ARM-based models, dLLM does not provide a convenient factorization that would allow us to compute likelihoods exactly. For efficiency reasons, we therefore must adopt additional approximations. Addressing this limitation is an important direction for future work.

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

# Appendix

## A  LLM USAGES.

Large Language Models (LLMs) were used solely for polishing the writing and improving the clarity of presentation. All ideas, analyses, results, and conclusions are original contributions of the authors.

## B  THEORETICAL ANALYSIS

In this section, we provide a theoretical foundation for our work. The central insight is that if the underlying hierarchical structure of reasoning can be learned from data, then there is a principled basis for designing algorithms that explicitly seek this structure. We establish this by showing that the latent concepts at each level of our proposed hierarchy are identifiable up to benign ambiguities.

### B.1  IDENTIFIABILITY OF LATENT CONCEPTS FROM OBSERVATIONS

Consider any level $l$ in the hierarchy. The concepts $\mathbf{S}_{l+1}$ at the next level are sampled based on $\mathbf{S}_l$ via a generating function $\mathbf{S}_{l+1} = f_{\mathbf{S}_{l+1}}(\mathbf{S}_l, \boldsymbol{\epsilon}_l)$, where $\boldsymbol{\epsilon}_l$ denotes the exogenous variables injected into level $l + 1$, independent of $\mathbf{S}_l$ and all variables at higher levels. The final response $\mathbf{R}$ can be expressed as $\mathbf{S}_l$ and the collection of exogenous variables $\mathbf{E}_l := (\boldsymbol{\epsilon}_m)_{m=l}^{L}$ via an invertible function $\mathbf{R} = q_l(\mathbf{S}_l, \mathbf{E}_l)$.

The following lemma shows how the hierarchically-dependent latent concepts $\mathbf{S}_{l+1}$ can be disentangled from the independent exogenous variables $\mathbf{E}_{l+1}$ for any $0 \le l < L$. The proof is inspired by previous work (Hyvarinen & Morioka, 2016; Hyvarinen et al., 2019; Kong et al., 2022).

**Lemma B.1** (Single-level Subspace Identifiability). *Assume the following data-generating process at a fixed, arbitrary $0 \le l < L$:*

$$\mathbf{S}_{l+1} \sim \mathbb{P}\left[\mathbf{S}_{l+1}|\mathbf{S}_l\right], \; \mathbf{E}_{l+1} \sim \mathbb{P}\left[\mathbf{E}_{l+1}\right], \; \mathbf{R} := q_{l+1}(\mathbf{S}_{l+1}, \mathbf{E}_{l+1}). \tag{5}$$

*We have the following conditions.*

  *i **Informativeness**: The function $q_{l+1}(\cdot)$ is a diffeomorphism.*

  *ii **Smooth Density**: The probability density function $p(\mathbf{S}_{l+1}, \mathbf{E}_{l+1}|\mathbf{S}_l)$ is smooth.*

  *iii **Sufficient Variability**: At any value $\mathbf{S}_{l+1}$, there exist $n(\mathbf{S}_{l+1}) + 1$ distinct values of $\mathbf{S}_l$, denoted as $\{\mathbf{S}_l^{(n)}\}_{n=0}^{n(\mathbf{S}_{l+1})}$, such that the vectors $\mathbf{w}(\mathbf{S}_{l+1}, \mathbf{S}_l^n) - \mathbf{w}(\mathbf{S}_{l+1}, \mathbf{S}_l^0)$ are linearly independent where $\mathbf{w}(\mathbf{S}_{l+1}, \mathbf{S}_l) = \left(\frac{\partial \log p(\mathbf{S}_{l+1}|\mathbf{S}_l)}{\partial \mathbf{S}_{l+1,1}}, \dots, \frac{\partial \log p(\mathbf{S}_{l+1}|\mathbf{S}_l)}{\partial \mathbf{S}_{l+1,n(\mathbf{S}_{l+1})}}.\right)$*

*If a model $\boldsymbol{\theta}$ satisfies i,ii, and iii, another model $\hat{\boldsymbol{\theta}}$ satisfies i,ii, and they generate identical distributions $\mathbb{P}\left[\mathbf{R}|\mathbf{S}_l\right] = \hat{\mathbb{P}}[\mathbf{R}|\mathbf{S}_l]$, then the latent concepts $\mathbf{S}_{l+1}$ are identifiable up to an invertible map, disentangled from $\mathbf{E}_{l+1}$: there exists an invertible mapping $\mathbf{S}_{l+1} \mapsto \hat{\mathbf{S}}_{l+1}$ where $\mathbf{S}_{l+1}$ and $\hat{\mathbf{S}}_{l+1}$ are generated in model $\boldsymbol{\theta}$ and $\hat{\boldsymbol{\theta}}$ respectively.*

*Proof.* Since we have matched distributions, it follows that:

$$p(\mathbf{R}|\mathbf{S}_l) = \hat{p}(\mathbf{R}|\mathbf{S}_l). \tag{6}$$

As the generating function $q_{l+1}$ has a smooth inverse (i), we can derive:

$$p(q_{l+1}(\mathbf{S}_{l+1}, \mathbf{E}_{l+1})|\mathbf{S}_l) = \hat{p}(\hat{q}_{l+1}(\hat{\mathbf{S}}_{l+1}, \hat{\mathbf{E}}_{l+1})|\mathbf{S}_l) \implies$$

$$p(\mathbf{S}_{l+1}, \mathbf{E}_{l+1}|\mathbf{S}_l)\left|\mathbf{J}_{q_{l+1}^{-1}}\right| = \hat{p}(q_{l+1}^{-1} \circ \hat{q}_{l+1}(\hat{\mathbf{S}}_{l+1}, \hat{\mathbf{E}}_{l+1})|\mathbf{S}_l)\left|\mathbf{J}_{q_{l+1}^{-1}}\right|.$$

Notice that the Jacobian determinant $\left|\mathbf{J}_{q_{l+1}^{-1}}\right| > 0$ because of $q_{l+1}(\cdot)$'s invertibility and let $h := q_{l+1}^{-1} \circ \hat{q}_{l+1} : (\hat{\mathbf{S}}_{l+1}, \hat{\mathbf{E}}_{l+1}) \mapsto (\mathbf{S}_{l+1}, \mathbf{E}_{l+1})$ which is smooth and has a smooth inverse thanks to those properties of $q_{l+1}$ and $\hat{q}_{l+1}$. It follows that

$$p(\mathbf{S}_{l+1}, \mathbf{E}_{l+1}|\mathbf{S}_l) = \hat{p}(h(\hat{\mathbf{S}}_{l+1}, \hat{\mathbf{E}}_{l+1})|\mathbf{S}_l) \implies$$

$$p(\mathbf{S}_{l+1}, \mathbf{E}_{l+1}|\mathbf{S}_l) = \hat{p}(\hat{\mathbf{S}}_{l+1}, \hat{\mathbf{E}}_{l+1}|\mathbf{S}_l)\left|\mathbf{J}_{h^{-1}}\right|.$$

The independence relation in the generating process implies that

$$\log p(\mathbf{S}_{l+1}|\mathbf{S}_l) + \sum_{i\in[n(\mathbf{E}_{l+1})]} \log p(\mathbf{E}_{l+1,i}) = \log \hat{p}(\hat{\mathbf{S}}_{l+1}|\mathbf{S}_l) + \sum_{i\in[n(\hat{\mathbf{E}}_{l+1})]} \log \hat{p}(\hat{\mathbf{E}}_{l+1,i}) + \log|\mathbf{J}_{h^{-1}}|.$$

(7)

For any realization $\mathbf{S}_l^0$, we subtract (7) at any $\mathbf{S}_l \neq \mathbf{S}_l^0$ with that at $\mathbf{S}_l^0$:

$$\log p(\mathbf{S}_{l+1}|\mathbf{S}_l) - \log p(\mathbf{S}_{l+1}|\mathbf{S}_l^0) = \log \hat{p}(\hat{\mathbf{S}}_{l+1}|\mathbf{S}_l) - \log \hat{p}(\hat{\mathbf{S}}_{l+1}|\mathbf{S}_l^0).$$

(8)

Taking derivative w.r.t. $\hat{\mathbf{E}}_{l+1,j}$ for $j \in [n(\hat{\mathbf{E}}_{l+1})]$ yields:

$$\sum_{i\in[n(\mathbf{S}_{l+1})]} \frac{\partial}{\partial \mathbf{S}_{l+1,i}}(\log p(\mathbf{S}_{l+1}|\mathbf{S}_l) - \log p(\mathbf{S}_{l+1}|\mathbf{S}_l^0)) \cdot \frac{\partial \mathbf{S}_{l+1,i}}{\partial \hat{\mathbf{E}}_{l+1,j}} = 0.$$

(9)

The left-hand side zeros out because $\hat{\mathbf{S}}_{l+1}$ is not a function of $\hat{\mathbf{E}}_{l+1}$.

Condition iii ensures the existence of at least $n(\mathbf{S}_{l+1})$ such equations with $\mathbf{S}_l^1, \ldots, \mathbf{S}_l^{n(\mathbf{S}_{l+1})}$ that are linearly independent, constituting a full-rank linear system. Since the choice of $j \in [n(\mathbf{E}_{l+1})]$ is arbitrary. It follows that

$$\frac{\partial \mathbf{S}_{l+1,i}}{\partial \hat{\mathbf{E}}_{l+1,j}} = 0, \forall i \in [n(\mathbf{S}_{l+1})], j \in [n(\mathbf{E}_{l+1})].$$

(10)

Therefore, the Jacobian matrix $\mathbf{J}_h$ is of the following structure:

$$\mathbf{J}_h = \begin{bmatrix} \dfrac{\partial \mathbf{E}_{l+1}}{\partial \hat{\mathbf{E}}_{l+1}} & \dfrac{\partial \mathbf{E}_{l+1}}{\partial \hat{\mathbf{S}}_{l+1}} \\ \dfrac{\partial \mathbf{S}_{l+1}}{\partial \hat{\mathbf{E}}_{l+1}} & \dfrac{\partial \mathbf{S}_{l+1}}{\partial \hat{\mathbf{S}}_{l+1}} \end{bmatrix}.$$

(11)

(10) suggests that the block $\frac{\partial \mathbf{S}_{l+1}}{\partial \hat{\mathbf{E}}_{l+1}} = 0$. Since $\mathbf{J}_h$ is full-rank, we can deduce that $\frac{\partial \mathbf{S}_{l+1}}{\partial \hat{\mathbf{S}}_{l+1}}$ must have full row-rank and $n(\mathbf{S}_{l+1}) \leq n(\hat{\mathbf{S}}_{l+1})$. Assuming the dimensions of the latent spaces are equal, $n(\mathbf{S}_{l+1}) = n(\hat{\mathbf{S}}_{l+1})$. Moreover, since $\mathbf{J}_h$ is full-rank and the block $\frac{\partial \mathbf{S}_{l+1}}{\partial \hat{\mathbf{E}}_{l+1}}$ is zero, we can derive that the corresponding block $\frac{\partial \hat{\mathbf{S}}_{l+1}}{\partial \mathbf{E}_{l+1}}$ in its inverse matrix $\mathbf{J}_{h^{-1}}$ is also zero. Therefore, there exists an invertible map $\mathbf{S}_{l+1} \mapsto \hat{\mathbf{S}}_{l+1}$, which concludes the proof. $\square$

With Lemma B.1 in hand, we can prove the following lemma that refines subspace invertible mappings $\mathbf{S}_{l+1} \mapsto \hat{\mathbf{S}}_{l+1}$ into component-wise invertible mappings $\mathbf{S}_{l+1,i} \mapsto \hat{\mathbf{S}}_{l+1,\hat{i}}$. That is, one can identify single dimensions on the level $l$.

To formalize our theoretical results, we introduce the following notation. For a matrix $M$, we denote its $i$-th row and $j$-th column as $M_{i,\cdot}$ and $M_{\cdot,j}$ respectively. We use $\cdot$ to indicate all the indices in that dimension. Recall the definition $\mathbf{S}_{l+1} := f_{l+1}(\mathbf{S}_l, \epsilon_l)$ and $\mathbf{R} := q_{l+1}(\mathbf{S}_{l+1}, \mathbf{E}_{l+1})$. We denote $D_{\mathbf{S}_l} f_{\mathbf{S}_{l+1}}$ as the partial derivative of the function $f_{\mathbf{S}_{l+1}}$ with respect to the higher-level variables $\mathbf{S}_l$. Let T be an arbitrary, fixed matrix with the same support as the matrix-valued function $\mathbf{T}(\cdot)$ in the relationship between two models' Jacobians: $D_{\mathbf{S}_l} \hat{f}_{\mathbf{S}_{l+1}} = \mathbf{T} D_{\mathbf{S}_l} f_{\mathbf{S}_{l+1}}$. Given a subset of indices $\mathcal{S} \subseteq \{1, \ldots, n\}$, we define the subspace $\mathbb{R}_{\mathcal{S}}^n$ as $\{s \in \mathbb{R}^n \mid s_i = 0 \text{ if } i \notin \mathcal{S}\}$. The support of the generative process for level $l + 1$ is defined as $\mathcal{D}_l := \text{supp}(D_{\mathbf{S}_l} f_{\mathbf{S}_{l+1}})$. The dependency structure is captured by a binary matrix $M_l$, where $M_{l,ij} = 1$ if and only if $(i, j) \in \mathcal{D}_l$. Let $\mathcal{A}_k$ be the set of indices for variables in $\mathbf{S}_{l+1}$ that depend on the higher-level variable $\mathbf{S}_{l,k}$. Let $d(\mathbf{S}_l)$ represent the dimensionality of $\mathbf{S}_l$. The following conditions follow prior work Zheng et al. (2025; 2022).

**Assumption B.2** (Non-degenerative Subspace Zheng et al. (2022)). *Suppose two alternative models $\boldsymbol{\theta}$ and $\hat{\boldsymbol{\theta}}$, with an $\ell_0$ regularization on $D_{\mathbf{S}_l} \hat{f}_{\mathbf{S}_{l+1}}$ such that $|\hat{\mathcal{D}}_l| \leq |\mathcal{D}_l|$, there exists a set of points $\{(\mathbf{S}_l, \theta)^{(\ell)}\}_{\ell=1}^{|\mathcal{D}_{l,\cdot,i}|}$ for each $\mathbf{S}_{l+1,i}$, such that:*

*1. The vectors $\{D_{\mathbf{S}_l} f_{\mathbf{S}_{l+1}}((\mathbf{S}_l, \theta)^{(\ell)}))_{\cdot,i}\}_{\ell=1}^{|\mathcal{D}_{l,\cdot,i}|}$ are linearly independent.*

2. *The transformed vectors lie in a subspace:* $\left[\mathrm{T}D_{\mathbf{S}_l} f_{\mathbf{S}_{l+1}}((\mathbf{S}_l, \theta)^{(\ell)})\right]_{\cdot,i} \in \mathbb{R}^{n(\mathbf{S}_{l+1})}_{\hat{\mathcal{D}}_{l,\cdot,i}}.$

We adapt a theoretical result from Zheng et al. (2025) as the following lemma.

**Lemma B.3** (Pair-wise Identification (Zheng et al., 2025)). *Let* $\theta := \left(f_{\mathbf{S}_{l+1}}, q_{\mathbf{S}_{l+1}}\right)$ *and* $\hat{\theta} :$ $\left(\hat{f}_{\mathbf{S}_{l+1}}, \hat{q}_{\mathbf{S}_{l+1}}\right)$ *be two alternative models. Suppose* $\theta$ *satisfies Condition B.1-i,ii, and Condition B.2, and* $\hat{\theta}$ *satisfies Condition B.1-i,ii, and an* $\ell_0$ *constraint* $\min \left\|\operatorname{supp} D_{\hat{\mathbf{S}}_l} \hat{f}_{\mathbf{S}_{l+1}}\right\|_0$. *If* $\theta$ *and* $\hat{\theta}$ *are observationally equivalent, i.e.,* $\mathbb{P}\left[\mathbf{R}|\mathbf{S}_l\right] = \hat{\mathbb{P}}[\mathbf{R}|\mathbf{S}_l]$ *for all* $\mathbf{S}_l$. *Then, the Jacobian of the transformation between the latent spaces satisfies:*

$$\frac{\partial \hat{\mathbf{S}}_{l+1,\pi(\mathcal{A}_i \backslash \mathcal{A}_j)}}{\partial \mathbf{S}_{l+1,\mathcal{A}_j}} = \mathbf{0} \quad and \quad \frac{\partial \hat{\mathbf{S}}_{l+1,\pi(\mathcal{A}_j \backslash \mathcal{A}_i)}}{\partial \mathbf{S}_{l+1,\mathcal{A}_i}} = \mathbf{0}, \tag{12}$$

*where* $\pi$ *is a permutation of the variable indices.*

**Assumption B.4** (Structural Diversity (Zheng et al., 2025)). *For any index* $i$ *of the variable* $\mathbf{S}_{l+1}$, *there exists a nonempty index set* $J$ *and a specific index* $j \in J$ *for* $\mathbf{S}_l$ *such that* $i$ *is the unique index in* $\mathcal{A}_j$ *that satisfies* $\{i\} = \mathcal{A}_j \setminus \cup_{k \in J \setminus \{j\}} \mathcal{A}_k$. *Moreover, the union* $\cup_{k \in J} \mathcal{A}_k$ *is equal to the entire index space* $[d(\mathbf{S}_l)]$.

**Lemma B.5** (Single-level Component-wise Identifiability). *Let* $\theta := \left(f_{\mathbf{S}_{l+1}}, q_{\mathbf{S}_{l+1}}\right)$ *and* $\hat{\theta} :$ $\left(\hat{f}_{\mathbf{S}_{l+1}}, \hat{q}_{\mathbf{S}_{l+1}}\right)$ *be two alternative models. Suppose* $\theta$ *satisfies Condition B.1-i,ii,iii and Condition B.4, and* $\hat{\theta}$ *satisfies Condition B.1-i,ii,iii, and a constraint on the support cardinality* $\min |\mathcal{D}_{\hat{\mathbf{S}}_l} \hat{f}_{\mathbf{S}_{l+1}}|$. *If* $\theta$ *and* $\hat{\theta}$ *are observationally equivalent, i.e.,* $\mathbb{P}[\mathbf{R}|\mathbf{S}_l] = \hat{\mathbb{P}}[\mathbf{R}|\mathbf{S}_l]$ *for all* $\mathbf{S}_l$, *then the variables* $\mathbf{S}_{l+1}$ *and* $\hat{\mathbf{S}}_{l+1}$ *are identifiable up to permutations and invertible transformations. Specifically, for any index* $i$, *there exists an invertible mapping* $S_{l+1,i} \mapsto \hat{S}_{l+1,\pi(i)}$ *for a permutation* $\pi$.

*Proof.* Notice that we have assumed all conditions for Lemma B.1 and Lemma B.3.

For any variable index $i$ on the level $l + 1$, invoking Lemma B.1 yields that

$$\frac{\partial \hat{S}_{l+1,\pi(i)}}{\partial \mathbf{E}_{l+1}} = 0. \tag{13}$$

Assumption B.4 suggests the existence of an index $j$ and an index set $J$ ($j \in J$) for the variable $\mathbf{S}_l$, such that the intersection such that $i$ is the only index in $\mathcal{A}_j$ that is unique to $\mathcal{A}_j$ (relative to other index sets $\{\mathcal{A}_k\}_{k \in J, k \neq j}$). Lemma B.3 implies that

$$\frac{\partial \hat{S}_{l+1,\pi(i)}}{\partial \mathbf{S}_{l+1,\cup_{k \in J} \mathcal{A}_k \backslash \{i\}}} = 0. \tag{14}$$

Moreover, since $\cup_{k \in J} \mathcal{A}_k = [d(\mathbf{S}_{l+1})]$ (Assumption B.4), we can deduce that

$$\frac{\partial \hat{S}_{l+1,\pi(i)}}{\partial \mathbf{S}_{l+1,[d(\mathbf{S}_{l+1})] \backslash \{i\}}} = 0. \tag{15}$$

Combining (13) and (15) yields

$$\frac{\partial \hat{S}_{l+1,\pi(i)}}{\partial \mathbf{S}_{l+1,[d(\mathbf{S}_{l+1})] \cup [d(\mathbf{E}_{l+1})] \backslash \{i\}}} = 0. \tag{16}$$

Recall that the mapping $(\mathbf{S}_{l+1}, \mathbf{E}_{l+1}) \mapsto (\hat{\mathbf{S}}_{l+1}, \hat{\mathbf{E}}_{l+1})$ is invertible. We can deduce from (16) that the mapping $S_{l+1,i} \mapsto \hat{S}_{l+1,\pi(i)}$ is invertible. Since the choice of $i \in [d(\mathbf{S}_{l+1})]$ is arbitrary, we have arrived at the desired conclusion. $\qquad \square$

Now, we are ready to present the identifiability for the entire hierarchical model. For ease of exposition, we consider the question variables $\mathbf{Q}$ as the top-level variable $\mathbf{S}_1$.

**Theorem B.6** (Identifiability of the Reasoning Hierarchy). *Let $\boldsymbol{\theta} := \left(f_{\mathbf{S}_{l+1}}, q_{\mathbf{S}_{l+1}}\right)_{l\in[L]}$ and $\hat{\boldsymbol{\theta}} : \left(\hat{f}_{\mathbf{S}_{l+1}}, \hat{q}_{\mathbf{S}_{l+1}}\right)_{l\in[L]}$ be two alternative models. Suppose every two adjacent levels $\mathbf{S}_l$ and $\mathbf{S}_{l+1}$ from $\boldsymbol{\theta}$ satisfy Condition B.1-i,ii,iii and Condition B.4, and every two adjacent levels $\hat{\mathbf{S}}_l$ and $\hat{\mathbf{S}}_{l+1}$ from $\hat{\boldsymbol{\theta}}$ satisfy Condition B.1-i,ii,iii, and a constraint on the support cardinality $\min|\mathcal{D}_{\hat{\mathbf{S}}_l}\hat{f}_{\mathbf{S}_{l+1}}|$. If $\boldsymbol{\theta}$ and $\hat{\boldsymbol{\theta}}$ are observationally equivalent, i.e., $\mathbb{P}\left[\mathbf{R}|\mathbf{S}_1\right] = \hat{\mathbb{P}}[\mathbf{R}|\mathbf{S}_1]$, then the variables $\mathbf{S}_l$ and $\hat{\mathbf{S}}_l$ ($l > 1$) are identifiable up to permutations and invertible transformations. Specifically, for any index $l > 1$ and $i \in d(\mathbf{S}_l)$, there exists an invertible mapping $S_{l,i} \mapsto \hat{S}_{l,\pi_l(i)}$ for a permutation $\pi_l$.*

*Proof.* Our proof is inductive. Theorem B.5 shows that if the variables at level $l$ are identifiable, then those at the next level, $l + 1$, are also identifiable. Since the top-level $\mathbf{S}_1$ is given, i.e., the question $\mathbf{Q}$, we can derive that all the variables in the hierarchical model are identifiable up to permutation and invertible transformations. That is, for any index $l > 1$ and $i \in d(\mathbf{S}_l)$, there exists an invertible mapping $S_{l,i} \mapsto \hat{S}_{l,\pi_l(i)}$ for a level-specific permutation $\pi_l$. □

## C  IMPLEMENTATION

We build our model based on the code repository by diffu-GRPO (Zhao et al., 2025). We apply GRPO to LLaDA-8B-Instruct (Nie et al., 2025). Following diffu-GRPO, we generate 6 rollouts per problem with a temperature of 0.9 and perform 12 update iterations per step (for Sudoku, we follow diffu-GRPO and use a temperature of 0.3 with 8 iterations). The model is trained with LoRA of rank 128 in 4-bit precision and evaluated in float16 precision. The learning rate is set to $3 \times 10^{-6}$ with 600 warm-up steps. During evaluation, we use zero-shot prompting and greedy decoding, with generation lengths of 128, 256, and 512 tokens, consistent with diffu-GRPO.

In order to compute the step-aware reward, we randomly select a timestep during the generation of rollouts for optimization. At this point, we take the intermediate generations consisting only of the text tokens that have been produced so far. We then concatenate 64 additional mask tokens to this partial sequence and feed the extended input back into the model. The model continues the process by performing iterative denoising based on this new input. Empirically, we find that using 64 mask tokens provides an effective trade-off between efficiency and performance on the benchmark datasets. After obtaining the outputs, we first compute the accuracy of the original rollouts, which reflects the correctness of the answers generated from the question alone. Next, we compute the accuracy of the newly generated answers obtained from the intermediate generations. Finally, we define the step-aware reward as the difference between these two accuracies, which quantifies the contribution of the intermediate generations to the final outcome.

## D  ADDITIONAL EXAMPLES

Here are additional responses comparisons for COUNTDOWN dataset.

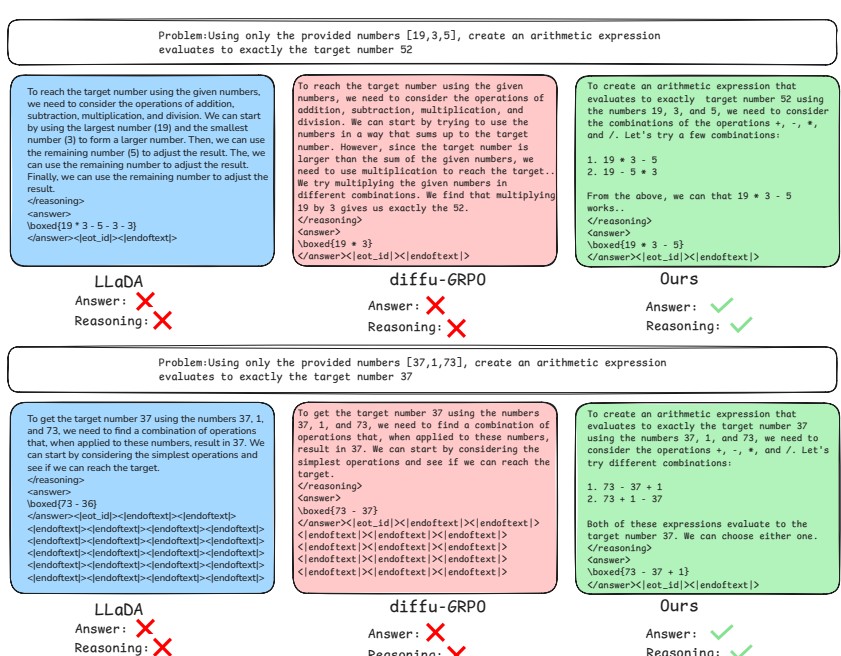

Figure 10: Comparison of generated responses across models. We observe that LLaDA produces meaningless reasoning steps, while diffu-GRPO generates incorrect reasoning. In contrast, our model correctly answers the question.

