# OpenReview forum: "Step-Aware Policy Optimization for Reasoning in Diffusion Large Language Models"
_ICLR.cc/2026/Conference — ICLR 2026 Conference Withdrawn Submission_

### Official Review · Reviewer_HnF3 · 2025-11-01

**Soundness:** 3
**Presentation:** 1
**Contribution:** 3
**Rating:** 6
**Confidence:** 4

**Summary:**

This work proposes an extension to diffuGRPO to encourage the model to make progress towards the correct answer throughout the sampling process. This work essentially develops an algorithm to automatically obtain process rewards. The process reward is computed as the difference between the average reward of rollouts from some intermediate step and rollouts from the initial masked sequence. This enables developing a process reward model solely from the verifiable outcome supervision. They validate their proposed algorithm across math benchmarks and synthetic reasoning puzzles and observe that it outperforms diffu-GRPO which only rewards the final rollout state.

**Strengths:**

This paper focuses on an area with a lot of interest. Strong DLLMs are being developed for the first time and it’s therefore of interest to figure out how to best replicate AR post-training procedures for DLLMs.

This is a natural, well-motivated way to develop process reward models for discrete diffusion models. The one-step probability estimation in diffu-GRPO likely fails to appropriately reward sampling trajectories in the same way as AR models and this might help address that.

There are some details that need to be figured out for the efficient application of their method. Namely fixing the baseline to be the full masked sequence and applying the indicator to the process reward are nice additions to make the algorithm tractable and effective.

The experimental evaluation is sound and their results are generally quite strong compared to diffu-GRPO across settings. The plots of the reward curves provide nice additional validation. I also appreciate the sampling step sweep in Table 3 to display the results across inference budgets.

**Weaknesses:**

The motivation talking about the “hierarchical decomposition of complex reasoning” feels quite spurious. At best, this may provide a conceptual motivation for the approach. While the theoretical framework may be interesting, it does not derive the algorithm. Claims like "this framework provides a principled foundation for algorithm design" and "motivated by this theory" overstate the connection.

In general, I find the theoretical framework to significantly detract from clarity. Process reward models are widely studied to provide intermediate supervision to LLMs (and are discussed in the related works section). This work proposes an algorithm to get automatic process rewards for discrete diffusion models. This is an interesting contribution in and of itself and the motivation for such work is very clear. For instance, the method is very similar to Math-Shepherd (which is cited) for AR methods.
This work inherits the mean-field assumption from diffu-GRPO which likely limits the effectiveness of the RL supervision.

The authors mention “Although MdLLMs offer faster inference compared to ARMs” (L 287). Although this is widely stated in the literature, this is typically not true. DLLMs with typical decoding settings are significantly slower than similarly sized ARMs due to the lack of kv-caching. For instance, see Fast-dLLM (Wu et al. 2025) for an inference-time comparison.

My understanding is that SAPO requires additional rollouts from the intermediate state. However, There is limited discussion/reporting of the computational cost compared to diffu-GRPO.

**Questions:**

How does the computational cost compare between SAPO and diffu-GRPO?

---

> ### Author Response · Authors · 2025-12-04
> **Response to Reviewer HnF3**
>
> Thank you for your constructive critique and we fully agree with your assessment.
> Prompted by your feedback, we have performed a major revision of the manuscript. We have rewritten the Abstract, Introduction, and Method sections to center the empirical contribution (SAPO) and moved the theory to a supporting role. Below, we detail how your specific comments drove these changes.
>
> >Q1. "The motivation talking about the 'hierarchical decomposition of complex reasoning' feels quite spurious... Claims like 'this framework provides a principled foundation...' overstate the connection."
>
> Thank you for the informative response. We are delighted that you find our empirical approach interesting and well-motivated. In light of your comments, we have revised the paper as follows.
> - **Rewritten Abstract**: We have completely removed the "theoretical framework" focus from the opening of the abstract. The new abstract now immediately introduces the practical problem (unstructured refinement) and the solution (SAPO).
> Old: _"We first propose a theoretical framework..."_
> New: _"In this work, we propose Step-Aware Policy Optimization (SAPO), a method to derive automatic process rewards for dLLMs without external supervision."_
> - **Revised Introduction**: We stripped out the heavy theoretical claims in Section 1. The introduction now focuses on the parallel with Process Reward Models (PRMs) in autoregressive models and the specific failure mode of "unstructured refinement" in diffusion models.
> - **Theory Demoted**: We removed the original Section 3 (A Hierarchical Formulation...) entirely. The conceptual framework is now condensed into Section 3.3, framed strictly as an interpretative lens ("Complexity Distribution") rather than a derivation.
> - **Methodological Clarity**: By removing the "Hierarchical Formulation" section, we now jump straight into the algorithm in Section 3. This allows readers to understand the mechanism (Equation 2) immediately without wading through definitions of latent selection variables.
> - **Explicit Framing**: We have updated the Related Work to explicitly position our method alongside automatic process reward extraction techniques like Math-Shepherd, highlighting that we extend this intuition to the non-autoregressive domain.
>
> > Q2. "This work inherits the mean-field assumption from diffu-GRPO which likely limits the effectiveness of the RL supervision."
>
> We appreciate you highlighting this constraint. We agree that the independence assumption in the reward estimation (inherited from diffu-GRPO) is a limitation. To address your concern, we have explicitly added a "Limitations" paragraph in Section 5, acknowledging that _"Our method relies on the mean-field assumption used in diffu-GRPO to estimate the log-likelihood of generated responses, which inherently neglects token-level dependencies. Unfortunately, this assumption is difficult to remove because, unlike ARM-based models, dLLM does not provide a convenient factorization that would allow us to compute likelihoods exactly. For efficiency reasons, we therefore must adopt additional approximations. Addressing this limitation is an important direction for future work.”_.
> That said, we appreciate the reviewer’s insight and would like to clarify that, although valuable, exact log-likelihood estimation falls somewhat outside the main focus of our method, which is directed toward enhancing reasoning quality.

---

> ### Author Response · Authors · 2025-12-04
>
> >Q3. "The authors mention 'Although MdLLMs offer faster inference compared to ARMs'... this is typically not true. dLLMs with typical decoding settings are significantly slower... due to the lack of kv-caching. For instance, see Fast-dLLM (Wu et al. 2025)..."
>
> We completely agree with this correction. We acknowledge that while dLLMs offer a theoretical reduction in the number of decoding steps (parallel generation), the wall-clock time is often slower in standard implementations due to the re-encoding overhead and the lack of native KV-caching support compared to highly optimized ARMs. To address your concern, we have done the following.
>
> - **Removed the Claim**: We have deleted the sentence "Although MdLLMs offer faster inference compared to ARMs" from the "Efficient Reward Estimation" paragraph in the Method Section. In the revised manuscript, this section now begins directly with the computational reality: "Generating multiple rollouts from intermediate states is expensive."
> - **Nuanced Discussion & Citation**: We have updated the Related Work (Section 2) to explicitly acknowledge the efficiency challenges and the necessity of recent innovations like KV-caching for dLLMs. We have included the citation you suggested, Fast-dLLM (Wu et al., 2025), along with others (e.g., Sparse-dLLM, dKV-cache), to correctly characterize the current state of inference efficiency
>
> >Q4. "How does the computational cost compare between SAPO and diffu-GRPO?"
>
>
> Thank you for your insightful question. We have added the speed comparison results to Table 2 in the revised manuscript. As shown below, our method introduces only a small training-time overhead compared to diffu-GRPO, while achieving substantially higher accuracy (an improvement of 55.4% on COUNTDOWN).
>
> | Model / Seq Len | 128 | 256 | 512 | sec/it |
> |-----------------|-----|-----|-----|--------|
> | **COUNTDOWN**   |     |     |     |        |
> | diffu-GRPO        | 33.2 | 31.3 | 37.1 | 3.19 |
> | Ours              | 51.6 | 52.0 | 56.3 | 3.42 |

---

### Official Review · Reviewer_nuUd · 2025-11-02

**Soundness:** 2
**Presentation:** 2
**Contribution:** 2
**Rating:** 4
**Confidence:** 5

**Summary:**

This paper proposes Step-Aware Policy Optimization (SAPO), an RL algorithm for diffusion LLMs that uses process-based rewards aligned with a theorized latent reasoning hierarchy. The authors formalize complex problem solving as a hierarchical selection process and argue that standard methods suffer from unstructured refinement—iterations that do not add meaningful progress. SAPO aims to guide denoising steps toward structured, coherent reasoning and reports gains on challenging reasoning benchmarks with improved interpretability.

**Strengths:**

1. Introduces a process-based reward tailored to dLLMs’ iterative denoising, rather than only outcome-based signals.

2. Provides a theoretical framework (hierarchical selection with identifiability insights) that motivates the algorithmic design.

**Weaknesses:**

1. The unstructured refinement claim ("identifies unstructured refinement as a key failure mode in standard diffusion language models, where the iterative denoising process fails to align with this latent reasoning hierarchy") is primarily theoretical and shown via case study, but lacks strong experimental validation. Prior MDM results (e.g., easy-first decoding) suggest hierarchical behaviors already exist to some extent [1][2]; the paper does not numerically define or analyze the latent hierarchy to test this claim.

2. The proposed process reward differs from AR LLM process supervision which splits the reasoning process sequentially in cot chain. Here, an intermediate sample $x_{t1}$​ is a full sequence. Thus, rewarding accuracy at $t_1$ may mainly encourage early stopping (that is to say when model can generate $x_{t1}$​ correctly before using all steps, this means the model is quite confident in this generation and can do early exit), and the model is rewarded based on the confidence and probably not from learning a multi-step latent hierarchy. This weakens the connection between the theory and the method design.

[1] https://arxiv.org/pdf/2308.12219v2
[2] https://arxiv.org/abs/2510.08632

**Questions:**

The citation format is incorrect somewhere e.g., lines 118–119.

---

> ### Author Response · Authors · 2025-12-04
> **Response to Reviewer nuUd**
>
> Thank you for your valuable comments. In light of your insightful feedbacks, we have conducted additional experiments and incorporated clarifications into the revised draft, which we believe is now much clearer and more convincing. Below, we provide our responses to your questions one by one.
>
>
> >Q1:"...numerically define or analyze the latent hierarchy to test this claim"
>
>
> In light of  your valuable suggestion. We have conducted a quantitative analysis of the latent hierarchy. Specifically, to empirically verify this prediction, we examine the causal structure induced by the learned models. At each prediction step, we estimate the causal links between hierarchy levels using token prediction probabilities. For example, given a partially decoded sequence such as "I am [mask]", we count how many vocabulary tokens have prediction probability greater than a small threshold (e.g., 0.01), treating each such token as an active causal link. We then compute the average and standard deviation of the number of causal links across timesteps.
>
> As shown in Table 3, our approach consistently produces fewer causal links than diffu-GRPO, indicating a sparser learned hierarchy. For instance, on COUNTDOWN, the number of causal links decreases from 4.37 to 3.80 (a 13.1\% reduction), and on SUDOKU it decreases from 4.31 to 3.90 (a 9.5\% reduction). Likewise, our model shows smaller standard deviations—e.g., from 2.41 to 2.04 on COUNTDOWN—demonstrating that the changes across hierarchy levels are smoother and more stable. Similar trends hold across GSM8K and MATH. These empirical findings closely align with our theoretical motivation and confirm that our reward encourages a more structured reasoning process.
>
>
> | Dataset   | diffu-GRPO      | Ours          |
> |-----------|------------------|---------------|
> | COUNTDOWN | 4.37±2.41        | 3.80±2.04     |
> | GSM8K     | 2.37±0.80        | 2.12±0.74     |
> | SUDOKU    | 4.31±2.95        | 3.90±2.44     |
> | MATH      | 3.19±1.21        | 3.11±1.24     |
>
>
>
> >Q2: "... an intermediate sample is a full sequence. Thus, rewarding accuracy at  may mainly encourage early stopping (that is to say when model can generate correctly before using all steps, this means the model is quite confident in this generation and can do early exit), and the model is rewarded based on the confidence and probably not from learning a multi-step latent hierarchy. This weakens the connection between the theory and the method design."
>
> We truly appreciate this thoughtful observation. In light of your comment, please let us clarify how we articulate the link between our method and the theoretical hierarchy.
>
> You are right that rewarding accuracy on intermediate samples encourages the model to reach the correct solution as early as possible. We view this behavior as a feature of the system rather than a conflict with the theory.
>
> Our theoretical framework models reasoning as a hierarchical selection process, but it does not mandate that every problem requires the same depth of decomposition. We see "early stopping" simply as the empirical realization of a **shallow hierarchy**.
> - **For simpler problems**: The uncertainty resolves quickly. The "incremental gain" in our reward (which calculates $Acc_{t_1} - Acc_{t_2}$) concentrates in the early stages. The model effectively learns that the latent structure for this specific input is shallow, allowing for an efficient early exit.
> - **For complex problems**: The model physically requires more denoising steps to resolve the high-entropy dependencies. In these cases, the accuracy gains are spread across the trajectory, incentivizing the model to engage the full, deep hierarchy.
>
> So, rather than bypassing the hierarchy, the model is actually adaptive: it dynamically matches the reasoning depth to the problem’s inherent complexity.
>
> Thank you again for helping us strengthen the connection between our efficiency results and our theoretical grounding.

---

### Official Review · Reviewer_Dwnt · 2025-11-03

**Soundness:** 2
**Presentation:** 3
**Contribution:** 2
**Rating:** 4
**Confidence:** 4

**Summary:**

This paper proposes step-aware policy optimization (SAPO) for training diffusion language models with RL. Their motivation is from the idea that complex reasoning should follow hierarchical decomposition rather than relying solely on outcome-based rewards. They introduce a Monte Carlo-based process reward that measures the contribution of intermediate denoising steps by comparing the final fully denoised completion's accuracy from different timesteps, providing richer learning signals than standard methods. The approach demonstrates improvements across six benchmarks with ablations on sampling parameters, generalization capability, and intermediate accuracy.

**Strengths:**

1. this paper presents motivation of why we should care about intermediate reasoning steps than out-come based learning in RLVR. and it proposes a MC estimate based method to quantify the advantage brought by sampling an intermediate step from t_2 to t_1. this brings more learning signal than merely just learn from the final fully denosied sequence.
2. this paper show better results in general as compared to previous methods across 6 benchmarks.
3. this paper has nice ablation studies on the number of N for the MC estimates, and generalization ability testing and intermediate answer comparison.
4. the paper's writing is clear and related work is comprehensive.

**Weaknesses:**

regarding the motivation of this paper:
1. The "we, we..." example is presented as evidence that the model lacks structured reasoning, but this is more like a coherence or quality issue. a model could generate coherent text while still having poor underlying reasoning structure. The paper assumes all reasoning problems require explicit hierarchical decomposition into discrete steps, but this doesn't account for simpler problems where reasoning might happen implicitly in the latent denoising process. Forcing explicit step-by-step reasoning on easy problems could be wasteful when the model might efficiently solve them through smooth constraint satisfaction in latent space. Their theory doesn't distinguish between problem complexity levels.

    notation wise: their model suggests R is generated from the final S_L but R could more naturally being a composition of all intermediate reasoning steps?
2. in figure 10, i wouldn't say generating repeated [eos] tokens are meaningless. LLaDA is pretrained with eos tokens as padding tokens, this is an expected behavior. It is intriguing to see model trained with SAPO doesn't generate repeated eos tokens after training. This means the model will always generate tokens to fill entire fixed generation space which seems unnatural.

Regarding experiments:

3. the performance improvement seems a little bit marginal as compared to diffu-grpo on gsm8k and math500 tasks in table 1.
4. Their motivation stems from structured decomposition of reasoning steps, yet their R_process reward is applied uniformly to all tokens rather than to specific reasoning steps. Additionally, their efficiency trick of setting t_2 = T (all masked tokens) means the reward granularity only measures progress from complete masking to t_1, which seems too coarse to capture the hierarchical step-by-step decomposition they theoretically motivate. There lacks an experiment on where t_2 not equal to T to align better with the motivation part in this paper in section 3.
5. Their method incurs increased computation by sampling N additional rollouts from intermediate state t_1 (roughly 50% more generations than diffu-GRPO). However, they provide no analysis of sample efficiency—comparing performance against total compute or number of generations.

**Questions:**

please see above.

minor problems:
1. figure 3 should have swapped the position of t2 and t1 in the right part of the figure?
2. Why only apply R_process to positive examples? If a response has both a wrong answer (A_i < 0) and poor intermediate reasoning (R_process < 0), will the model be discouraged more strongly?

---

> ### Author Response · Authors · 2025-12-04
> **Response to Reviewer Dwnt**
>
> Thank you for the thoughtful and constructive feedback. Your comments helped us identify a disconnect between our theoretical formulation and its practical interpretation in the manuscript.
> Below, we address your concerns point-by-point.
>
> >Q1. "The 'we, we...' example is presented as evidence that the model lacks structured reasoning, but this is more like a coherence or quality issue.":
>
> We completely agree. The original example conflates coherence with logical structure. We have revised the Introduction and Figure 1 to distinguish between coherence (grammatical fluency) and structured reasoning (logical progression). We now clarify that unstructured refinement refers to the generation of tokens—whether coherent or repetitive—that fail to reduce the problem's complexity or advance the solution state.
> 1. **Revision to the Introduction.** We have refined the definition of "unstructured refinement" to clarify that repetition is just one symptom of a lack of process supervision, distinguishing it from general coherence issues.
> **Original Text**: "We term this critical failure mode unstructured refinement: the model wastes most of its iterative steps on unproductive tokens..."
> **Revised Text**: "We term this critical failure mode unstructured refinement. While models may maintain local textual coherence, they often fail to utilize the iterative denoising process for logical progression. This results in the model wasting steps on unproductive tokens—manifesting as repetitive loops (mode collapse) or coherent but vacuous 'fluff'—forcing the final few steps to bridge the entire complexity gap."
> 2. **Revision to Figure 1 Caption.** We have updated the caption to acknowledge that the "we, we..." example represents a specific type of failure (mode collapse) that highlights the lack of incentive to reason.
> **Original Caption**: "Figure 1: The problem of unstructured refinement. A standard MdLLM trained with an outcome-only reward produces a correct answer but fills its reasoning trace with meaningless, repetitive tokens. This indicates the iterative process is not contributing meaningfully to the solution."
> **Revised Caption**: "Figure 1: The problem of unstructured refinement. A standard MdLLM trained with outcome-only rewards produces a correct answer but fills its reasoning trace with meaningless tokens. While this specific example exhibits mode collapse (a coherence failure), it serves as a stark illustration of a broader issue: the iterative process is not incentivized to reduce problem complexity, allowing the model to 'spin' on unproductive steps while coincidentally hitting the correct answer."
>
> It is also worth noting that we agree that, in some cases, the reasoning may occur entirely within the latent process. However, in many scenarios, we still need meaningful, human-interpretable reasoning so that people can verify and understand how the model arrives at its final answer. This is especially important in settings where the entire decision-making process must remain transparent.

---

> ### Author Response · Authors · 2025-12-04
>
> > Q2. "The paper assumes all reasoning problems require explicit hierarchical decomposition... this doesn't account for simpler problems where reasoning might happen implicitly... Forcing explicit step-by-step reasoning on easy problems could be wasteful... Notation wise: their model suggests R is generated from the final $S_L$ but R could more naturally be a composition of all intermediate reasoning steps?"
>
> We are very grateful for this insight. This is a really helpful observation, and it pushed us to clarify exactly how our theoretical variables ($S_l$) map onto the practical diffusion process. We entirely agree that forcing a rigid 'step-by-step' structure on trivial problems would be inefficient. That is why we designed our method to be flexible to accommodate these cases. it allows the denoising trajectory to naturally adapt to the problem’s inherent complexity rather than mandating a fixed logical depth
> - A. Clarifying $S_l$ as Denoising States (Addressing Notation)
> You are absolutely right that $R$ acts as a composition of steps. In our revised framework, we clarify that the levels $S_1, \dots, S_L$ do not represent sequential logical sentences, but rather the state of the sequence at varying levels of the diffusion denoising process.
> -- $S_0$ is the noisy/masked state.
> -- $S_L$ (which is $R$) is the fully denoised state.
> -- Therefore, the "hierarchical decomposition" is actually a resolution of uncertainty. $S_l$ captures the entire text sequence at a specific level of refinement.
>
> - B. Theoretical Expressiveness for Varying Complexity
> Prompted by your concern that our theory might not distinguish complexity levels, we have reinstated and expanded our discussion on the expressiveness of the hierarchical abstraction. We posit that the hierarchy represents the potential complexity of the reasoning space (the full graph).
> -- **For complex problems**: The model utilizes the full depth of the hierarchy to resolve dependencies.
> -- **For simple problems**: The problem activates only a **sparse subgraph** of constraints. In the context of diffusion, this means the transition between levels $p(S_{l+1}|S_l)$ becomes trivial (effectively an identity mapping where unmasked tokens are simply retained).
>
> We have added the following clarification to Section 3.3 to address this explicitly:
> _"We do not posit this hierarchical structure as a rigid, universal cognitive model. Rather, we propose the hierarchy as a flexible abstraction for the potential reasoning complexity. Simpler problems activate only a sparse subgraph of the available constraints. In the context of diffusion, this manifests as trivial transformations where the 'reasoning' happens implicitly via smooth constraint satisfaction in the latent space, without requiring complex structural decomposition."_
>
> >Q3. "In figure 10, i wouldn't say generating repeated [eos] tokens are meaningless... This means the model will always generate tokens to fill entire fixed generation space which seems unnatural."
>
> We agree that [EOS] tokens serve a functional padding role in pre-training. We want to empahsize that baseline approach generates some meaningless reasonings that do not contribute to the final answer. We have changed the caption: _“Comparison of generated responses across models. We observe that LLaDA produces meaningless reasoning steps, while diffu-GRPO generates incorrect reasoning. In contrast, our model correctly answers the question.”_
>
>
> >Q4. "..lacks an experiment on where t_2 not equal to T to align better with the motivation part in this paper in section 3."
>
> Thank you for your insightful comments. We agree that adding an experiment on this design choice better aligns with our motivation. Accordingly, we conducted an ablation study. In Ours-Cover, we compute the average reward across all timestep intervals, while Ours-Random selects $t_2$ randomly instead of fixing it to $T$. Both variants achieve strong performance on the COUNTDOWN dataset, but at the cost of slower training. Our proposed approach attains results close to Ours-Cover and Ours-Random, yet with substantially higher efficiency (3.42 sec/it vs. 6.23 and 4.76 sec/it, respectively).
>
> | Model / Seq Len | 128 | 256 | 512 | sec/it |
> |-----------------|-----|-----|-----|--------|
> | **COUNTDOWN**   |     |     |     |        |
> | diffu-GRPO        | 33.2 | 31.3 | 37.1 | 3.19 |
> | Ours-Cover        | 55.1 | 59.4 | 58.2 | 6.23 |
> | Ours-Random       | 55.4 | 54.7 | 59.8 | 4.76 |
> | Ours              | 51.6 | 52.0 | 56.3 | 3.42 |
>
> >Q5. Sample efficiency comparison.
>
> Thanks for your suggestion. As shown in the above table, our approach achieves 51.6 accuracy at a training speed of 3.42 sec/it, whereas the baseline diffu-GRPO reaches only 33.2 accuracy with 3.19 sec/it. By increasing the training cost only slightly, we obtain a 55.4% accuracy improvement over the baseline.

---

### Official Review · Reviewer_qcGf · 2025-11-05

**Soundness:** 2
**Presentation:** 2
**Contribution:** 2
**Rating:** 2
**Confidence:** 4

**Summary:**

The paper is concerned with post-training of diffusion language models (dLLMs). Specifically, they focus on complex reasoning, and argue that the use of sparse, outcome-based rewards
can lead to models producing correct answers via incorrect reasoning: what they label as "unstructured refinement". The authors propose a framework which considers the full reasoning chain as a latent hierarchical process. They then try to align the dLLM with latent hierarchy using RL through SAPO ( Step-Aware Policy Optimization ). SAPO involves using a process reward that can capture 'implicit progress' between two stages in the denoising path, and trying to align the denoising with the corresponding hierarchical reasoning structure. Empirical results on mathematical and logical reasoning benchmarks (GSM8K, MATH, etc.) are presented to show performance improvements.

**Strengths:**

Applying process-reward models to dLLMs is novel to the best of my knowledge. The paper is also generally well-structured and easy to follow. The problem of inducing hierarchical reasoning to solve the "unstructured refinement" problem is interesting. dLLM chains are in general harder to interpret and understand, and improving the reliability of the generation process is of practical value.

**Weaknesses:**

he theoretical construct for reasoning has weak links to the proposed method. It seems to have been forced in, instead of naturally translating to it. On the other hand the high level proposal is intuitive, and can be presented directly. The issues I see are: a) the framework assumes an identifiable process, but this has no analogue for the actual process,  b) sparsity is considered important for the identification but does not appear at all in the proposal, c) how does the random intervals of the denoising process connect to 'a set of logical constraints' from the theory.  Finally  and importantly I do not see how the model actually learns the postulated hierarchy.

Somewhat relatedly, we have the upweighting heuristics. The authors provide the process reward, only if the base advantage is positive ( or effectively the answer is correct). The authors justify this as avoiding boosting process chains which are 'correct/better' but go nowhere. This is prioritizing final answer accuracy over reasoning integrity, and is not punishing models that get the right answer for the wrong reason undermining the core thesis of the paper. Furthermore, the whole point of process models (in general) is to promote the model to do correct reasoning even on wrong chains, as these correct subchains will then be promoted and the model generalize better. This makes me consider that the process reward models are not learning something effective in the first place.

I am also confused by equation 4. Do you sample multiple random intervals and use the empirical average, or is it one sample only. The latter seems surprising, and if its an empirical average, Eq 4 seems wrong. I also think  a more structured sampling scheme would be important to check.

**Questions:**

I think the empirical evaluation is weak. The paper fails to demonstrate that SAPO provides a unique advantage over simply applying a similar process reward from a standard ARM. A good baseline might be to use a auto-regressive LLM process model, by considering the mask as sometype of space or pad tokens. I think SAPO should be compared against a strong baseline that uses an analogous process reward. It seems difficult to determine if the differences come from the dLLM based rewards or the process reward strategy.


The up-weighting strategy in Eq. 5 undermines the goal of eliminating "correct-by-chance" solutions. Can you run an ablation without the indicator function there and normal process reward.

The model seems to be worse than even d1/diffu-GRPO. Why would process reward models make the model worse? The difference on GSM are also small. Finally other models, which do not need  any process model etc. and have better results ( See [1])
Given the goal is the connection to process model, I think one should add an evaluation of the process reward model.

If I am understanding the reward over a single random interval correctly (and I may not be), is there a validation of this single random interval is an "effective and efficient approximation."



[1] wd1 Weighted Policy Optimization for Reasoning in Diffusion Language Models

---

> ### Author Response · Authors · 2025-12-04
> **Response to Reviewer qcGf**
>
> Thank you for your critical and thoughtful feedback. In light of your valuable feedback, we have come up with a comprehensive revision that re-centers the paper around its primary contribution: the methodology of extracting automatic progressive rewards from the diffusion process.
> We hope that this shift, inspired directly by your advice, has significantly improved the clarity and scientific grounding of the manuscript. Below, we address your concerns point-by-point and report the exact changes we have made.
>
> >Q1 “...there is a disconnect between the theoretical construct and the method... the high-level proposal is intuitive enough to stand on its own without a forced theoretical framing...”
>
> We are deeply grateful for this insight. Following your suggestion, we have restructured the narrative to prioritize the empirical contribution as a solution to the supervision scarcity problem in dLLMs. The theory is now presented as a tool for understanding why the method works, rather than the primary driver.
> **Specific Edits**:
> - **Introduction (Section 1)**: We have rewritten the introduction to focus on **“unstructured refinement”** as the core problem. We now frame SAPO as a method to derive automatic process supervision to prevent models from wasting diffusion steps, moving away from abstract theoretical claims.
> - **Methodology (Section 3)**: The description of the algorithm now precedes the theory. We detail the automatic reward extraction first, establishing the method's standalone value.
>
> >Q2. “...the link between the ‘sparsity constraint’ theory and the model learning process was weak... it wasn't clear how the model actually learns the postulated hierarchy...”
> >
> This is an excellent point. Instead of asserting that the model explicitly learns a hierarchy, we now use the theory to justify why our proposed reward function is effective. We clarify that the reward encourages a complexity distribution that lessens the burden for each step. Therefore, each step/function could be made simple (sparse).
> **Specific Edits:**
> - **New Subsection 3.3 (Theoretical Understanding)**: We compressed the theory section and moved it to the end of the Methodology as Section 3.3.
> - **Revised Wording**: We changed the language from "Motivation" to "Theoretical Understanding" and "Justification."
> - **Theorem Context**: We added text stating: "We interpret the benefits of intermediate rewards through the lens of complexity reduction... Intuitively, if a model can decompose a complex function into a composition of sparse, simple functions, it can more easily learn a natural, robust reasoning process."
> - **Empirical Evidence**: We have added quantitative experiments in Table 3 to demonstrate that our model learns a more structured hierarchical representation. For each timestep at which tokens are unmasked, we approximate the presence of causal links from the current tokens to tokens in previous timesteps by thresholding the predicted probability distribution. For example, given the partially decoded sequence “I am [mask],” we count the number of causal edges as the number of candidate tokens whose predicted probabilities for filling the [mask] exceed 0.01. We report the average and standard deviation of the number of such causal links across timesteps.
> As shown in Table 3, our model induces a sparser hierarchy compared to diffu-GRPO. On the COUNTDOWN dataset, diffu-GRPO yields an average of 4.37 causal links, whereas our method produces 3.80. Moreover, the standard deviation of our model’s causal link count is 2.04, substantially lower than diffu-GRPO’s 2.41, indicating that our method learns a more sparse and stable hierarchical structure with fewer abrupt variations over timesteps.
>
> | Dataset   | diffu-GRPO      | Ours          |
> |-----------|------------------|---------------|
> | COUNTDOWN | 4.37±2.41        | 3.80±2.04     |
> | GSM8K     | 2.37±0.80        | 2.12±0.74     |
> | SUDOKU    | 4.31±2.95        | 3.90±2.44     |
> | MATH      | 3.19±1.21        | 3.11±1.24     |

---

> > ### Author Response · Authors · 2025-12-04
> >
> > > Q3. “...provides a unique advantage over simply applying a similar process reward from a standard ARM”
> >
> > Thank you for your insightful comments. We agree that adding a comparison with the PRM from ARM would better highlight the advantages of our approach. To this end, we incorporated a recent PRM from ARM [1], using their released Mistral-7B PRM checkpoints for GSM8K and MATH. We inserted a special step-tag every 16 timesteps so the PRM could segment reasoning steps and computed the process reward as the average PRM score across these intervals. However, when applying the PRM for policy optimization in dLLM, we observed several limitations (see Figure 5 in the revised manuscript): (1) **Huge memory consumption**. Unlike our approach, which uses the training model itself to compute rewards, the pretrained PRM requires substantial GPU and CPU memory, resulting in significantly slower training (7.62 sec/it vs. 3.42 sec/it for ours). (2) **Instability**. Feeding model-generated responses into the PRM often produced NaN values, likely because the PRM expects highly structured inputs such as explicit “step1/step2” formatting; we therefore had to replace NaN scores with zero. (3) **Potential reward hacking**. Although the PRM reward increased during training, the final task performance did not improve, suggesting that the policy may be exploiting weaknesses in the PRM scoring model rather than improving actual reasoning quality—consistent with known reward-model exploitation phenomena. As reported in the revised Table 1, the PRM achieves 71.7, 80.9, and 81.5 on GSM8K, while diffu-GRPO obtains 72.6, 79.8, and 81.9, and our method further improves performance to 72.9, 82.2, and 82.4. These results highlight the advantages of our approach over standard PRM-based training.
> >
> >   | Model              | GSM8K @128 | GSM8K @256 | GSM8K @512 | MATH @128 | MATH @256 | MATH @512 |
> > |--------------------|------------|------------|------------|-----------|-----------|-----------|
> > | diffu-GRPO         | 72.6       | 79.8       | 81.9       | 33.2      | 37.2      | 39.2      |
> > | diffu-GRPO+PRM     | 71.7       | 80.9       | 81.5       | 30.8      | 36.0      | 36.0      |
> > | Ours               | 72.9       | 82.2       | 82.4       | 32.0      | 40.0      | 38.4      |
> >
> >
> >
> > [1] Zhang, H., Wang, P., Diao, S., Lin, Y., Pan, R., Dong, H., Zhang, D., Molchanov, P. and Zhang, T., Entropy-Regularized Process Reward Model, TMLR.
> >
> > > Q4. “...prioritizing final answer accuracy might reinforce 'right answer, wrong reason' behavior... heuristics may fail to punish flawed reasoning...”
> >
> > Thank you for the thoughtful insight. Our rewards prioritise samples with both correct answers and good reasoning steps. Please note that the final advantage is computed by adding the proposed reward; therefore, we are discouraging samples with bad reasoning and samples with only correct answers implicitly, since the gradient for optimization will be more influenced by samples with higher advantage. We totally agree with you that many process reward models are to “promote the model to do correct reasoning even on wrong chains,” but they are mostly deployed for test-time selection, and the final answers are unknown. In this paper, we propose a step-aware reward for policy optimization during training time. This up-weighting strategy could further leverage the information of answers to promote better reasoning. We have revised our text to make it clearer.
> >
> > **Empirical Evidence**: In light of your insightful suggestion, we conducted an ablation study by removing the up-weighting strategy—i.e., we computed the proposed reward uniformly for all samples. As shown in Table 2 of the revised draft, the baseline diffu-GRPO achieves scores of 33.2, 31.3, and 37.1 on the COUNTDOWN dataset, whereas our approach without up-weighting attains 41.0, 41.4, and 50.4, demonstrating the effectiveness of the proposed reward itself. In contrast, our full model with up-weighting achieves 51.6, 52.0, and 56.3. These results indicate that the up-weighting strategy further improves performance by preventing overly harsh penalties on samples that have correct answers but imperfect reasoning chains.
> >
> >
> > | Model / Seq Len   | 128  | 256  | 512  | sec/it |
> > |-------------------|------|------|------|--------|
> > | diffu-GRPO        | 33.2 | 31.3 | 37.1 | 3.19   |
> > | diffu-GRPO+PRM    | -    | -    | -    | 7.58   |
> > | Ours-NoUpweight   | 41.0 | 41.4 | 50.4 | 3.42   |
> > | Ours-Cover        | 55.1 | 59.4 | 58.2 | 6.23   |
> > | Ours-Random       | 55.4 | 54.7 | 59.8 | 4.76   |
> > | Ours              | 51.6 | 52.0 | 56.3 | 3.42   |

---

> > > ### Author Response · Authors · 2025-12-04
> > >
> > > > Q5. “...confusion regarding Equation 4... specifically whether it uses a single sample or an empirical average...”
> > >
> > > Thank you for your question. For each sample, we randomly select a timestep and compute the reward at that timestep as an approximation of the empirical average over all timesteps, primarily for efficiency reasons. We include an ablation study in Table 3: Ours-Cover corresponds to computing the empirical average over all timestep intervals, while Ours-Random computes the reward from a single randomly selected interval.
> > > We observe that both Ours-Cover (55.1, 59.4, 58.2) and Ours-Random (55.4, 54.7, 59.8) achieve strong performance—substantially outperforming the baseline diffu-GRPO. However, Ours-Cover is significantly slower because it requires computing the reward for every interval (6.23 sec/it), which is nearly twice the training time of our proposed method (3.42 sec/it). Ours-Random (4.76 sec/it) is also slower because it must compute completions at two timesteps for each sample, whereas our method requires only a single forward pass by reusing the t2=T completion for all reward components.
> > > Moreover, Ours-Random cannot leverage the final answer to filter samples and therefore must compute completions for all examples. In contrast, our approach achieves competitive performance (51.6, 52.0, 56.3) using only one model forward pass per sample. It is also much faster (3.42 sec/it), only slightly slower than diffu-GRPO (33.2, 31.3, 37.1; 3.19 sec/it), while delivering far superior accuracy.
> > > We are deeply grateful for your guidance, which has elevated the quality of this paper significantly.

---

### Author Response · Authors · 2025-12-04
**Summary of the Rebuttal**

**Dear Area Chair,**

Thank you for coordinating the ICLR review process. In response to the thoughtful critiques from Reviewers **qcGf**, **Dwnt**, **nuUd**, and **HnF3**, we have undertaken substantial revisions that clarify the paper’s contribution, strengthen empirical evidence, and refine theoretical positioning. Below we summarize the key issues and the concrete changes made in the revised manuscript.



| **Concern** | **Our Resolution** | **Supporting Evidence / Changes** |
|----------------------|--------------------|-----------------------------------|
| **(qcGf)** *Theory–method disconnect; sparsity & identifiability unclear; hierarchy seems unproven* | We restructured the paper so SAPO’s empirical method comes first and the theory serves only as a **supporting interpretation** rather than a derivation. | **(1)** Abstract & Introduction rewritten to emphasize automatic process reward extraction. **(2)** Removed the previous “Hierarchical Formulation” section; theory condensed into **Section 3.3 (Theoretical Understanding)**. **(3)** Language toned down (“interpretive lens,” not “principled foundation”). |
| **(qcGf & nuUd)** *Evidence to show SAPO induce a hierarchy* | Added a **new quantitative causal-link analysis** demonstrating that SAPO learns **sparser and more stable** dependency structures along the denoising trajectory. |  **(1)** **Table 3:** COUNTDOWN causal links: 4.37±2.41 → **3.80±2.04**; similar reductions on GSM8K, SUDOKU, MATH. **(2)** Confirms that intermediate rewards encourage structured refinement. |
| **(qcGf)** *Need baseline comparison to ARM PRMs* | Included a **strong PRM baseline** using the ARM Entropy-Regularized PRM (Mistral-7B), adapted for dLLMs. Demonstrated major stability and efficiency benefits of SAPO. | **(1)** **Table 1:** GSM8K @512 — diffu-GRPO: 81.9; PRM: 81.5; **SAPO: 82.4**. **(2)** Training cost: PRM = **7.58 sec/it** vs SAPO = **3.42 sec/it**. **(3)** PRM produced NaNs and signs of reward hacking. |
| **(qcGf)** *Need ablation on up-weighting.* | Added a **no-upweighting ablation** and clarified that the final advantage already incorporates reasoning quality. SAPO without up-weighting still surpasses diffu-GRPO; the heuristic further stabilizes training. | **(1)** **Table 2:** diffu-GRPO: 33.2 → **SAPO-NoUpweight: 41.0** → **SAPO: 51.6** (COUNTDOWN @128). **(2)** Shows the reward drives gains; up-weighting enhances sample efficiency without masking reasoning flaws. |
| **(qcGf & Dwnt)** *Equation 4 unclear; random intervals vs empirical average; $t_₂ = T$ may be too coarse.* | Clarified Eq. 4 implementation and added **interval ablations**: full averaging, random t₂, and SAPO’s $t_₂ = T$. SAPO offers the best efficiency–performance tradeoff. | **(1)** Cover: 55.1/59.4/58.2 at **6.23 sec/it**.• Random: 55.4/54.7/59.8 at **4.76 sec/it**. **(2)** **SAPO ($t_₂ = T$): 51.6/52.0/56.3 at 3.42 sec/it**. **(3)** SAPO reuses the final completion, explaining efficiency. |
| **(Dwnt)** *Unstructured-refinement example conﬂates coherence with reasoning structure.* | Revised Introduction & Figure 1 to separate coherence failures (e.g., repetition) from **failure to reduce problem complexity**. | **(1)** Updated text defines unstructured refinement as failure of *complexity reduction*, not mere incoherence. **(2)** Figure 1 caption rewritten accordingly. |
| **(Dwnt)** *Not all tasks need explicit multi-step hierarchy; notation unclear.* | Clarified that the hierarchy is a **flexible abstraction**: shallow for easy tasks, deep for hard ones. Updated notation: $S_l$ are denoising states, not literal natural-language steps. | **(1)** Section 3.3 expanded to explain adaptive reasoning depth. **(2)** Clarified $S_L = R$ as the full sequence composed from intermediate states. |
| **(HnF3)** *Theoretical framing detracts from clarity; mean-field assumption limits expressiveness; dLLM inference-speed remark incorrect; compute cost missing.* | Theory softened; limitations acknowledged; incorrect speed claim removed; compute cost added explicitly. | **(1)** Removed claims that theory “derives” algorithm.• Added Limitations paragraph noting dependence on diffu-GRPO’s mean-field assumption. **(2)** Deleted incorrect dLLM speed statement; updated Related Work to cite Fast-dLLM. **(3)** Added compute comparison table showing SAPO = **3.42 sec/it** vs diffu-GRPO = **3.19 sec/it**, with **55% accuracy improvement**. |

---


With the revised manuscript and the responses provided, we believe we have adequately addressed the reviewers’ concerns. We hope that our approach, which combines diffusion processes with progress-based rewards, can offer useful theoretical and practical insights and contribute to future developments in the community.

**Best regards,**
*The Authors*

---

### Note · Authors · 2026-01-06

I have read and agree with the venue's withdrawal policy on behalf of myself and my co-authors.